# Language-based Trial and Error Falls Behind in the Era of Experience

**Haoyu Wang** [1]  **Guozheng Ma** [1]  **Shugang Cui** [2]  **Yilun Kong** [1]  **Haotian Luo** [3]  **Li Shen** [4]  **Mengya Gao** [2]
**Yichao Wu** [2]  **Xiaogang Wang** [2]  **Dacheng Tao** ✉ [1]

## Abstract

While Large Language Models (LLMs) excel in language-based agentic tasks, their applicability to unseen, nonlinguistic environments (e.g., symbolic or spatial tasks) remains limited. Previous work (Chen et al., 2025) attributes this performance gap to the mismatch between the pretraining distribution and the testing distribution. In this work, we demonstrate the primary bottleneck is the prohibitive cost of exploration: mastering these tasks requires extensive trial-and-error, which is computationally unsustainable for parameter-heavy LLMs operating in a high dimensional semantic space. To address this, we propose **SCOUT** (**S**ub-**S**cale **C**ollaboration **O**n **U**nseen **T**asks), a novel framework that decouples exploration from exploitation. We employ lightweight "scouts" (e.g., small MLPs) to probe environmental dynamics at a speed and scale far exceeding LLMs. The collected trajectories are utilized to bootstrap the LLM via Supervised Fine-Tuning (SFT), followed by multi-turn Reinforcement Learning (RL) to activate its latent world knowledge. Empirically, SCOUT enables a Qwen2.5-3B-Instruct model to achieve an average score of 0.86, significantly outperforming proprietary models, including Gemini-2.5-Pro (0.60), while saving about 60% GPU hours consumption. The code is available at https://github.com/Harry-mic/SCOUT.

## 1. Introduction

Large Language Models (LLMs) have demonstrated remarkable capabilities across a wide range of tasks (Guo et al., 2025; Wang et al., 2025d; Zhang et al., 2025; Wang et al., 2025b; Yao et al., preprint; Shridhar et al., 2020), primarily

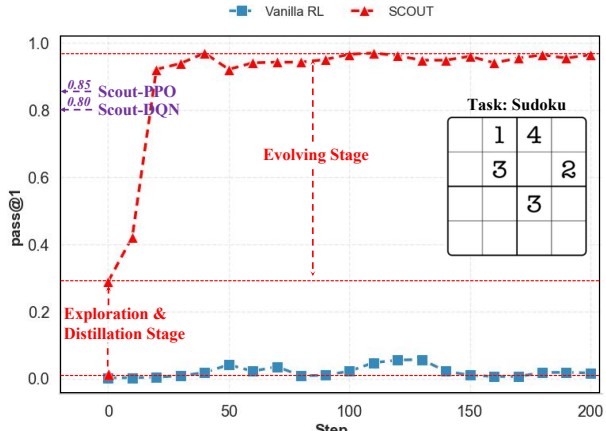

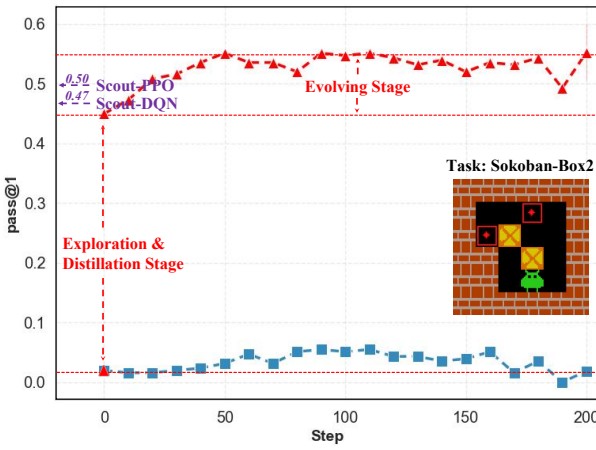

*Figure 1.* Exploration and Distillation Stage will either directly award the language models the skills, such as the performance on Sokoban-Box2 (below) that leads to direct convergence (0.0 to 0.45), or indirectly teach relevant knowledge that will be later activated via Evolving Stage, such as the performance on Sudoku (above figure, from 0.0, to 0.29, then to 0.97).

driven by extensive pretraining on high quality text corpora (Brown et al., 2020; Touvron et al., 2023; Yang et al., 2025; Zhou et al., 2023). This equips LLMs with broad world knowledge, enabling strong zero-shot generalization in language correlated scenarios such as creative writing, summarization, reasoning, and even language based agentic tasks (Li et al., 2023; Zheng et al., 2023; Shao et al., 2024; Shridhar et al., 2020; Yao et al., preprint). However, when deployed in unseen, non-linguistic tasks such as spa-

---

[1]Nanyang Technological University [2]SenseAuto [3]Sichuan University [4]Sun Yat-sen University. Correspondence to: Dacheng Tao (✉) <dacheng.tao@ntu.edu.sg>.

*Proceedings of the 43rd International Conference on Machine Learning*, Seoul, South Korea. PMLR 306, 2026. Copyright 2026 by the author(s).

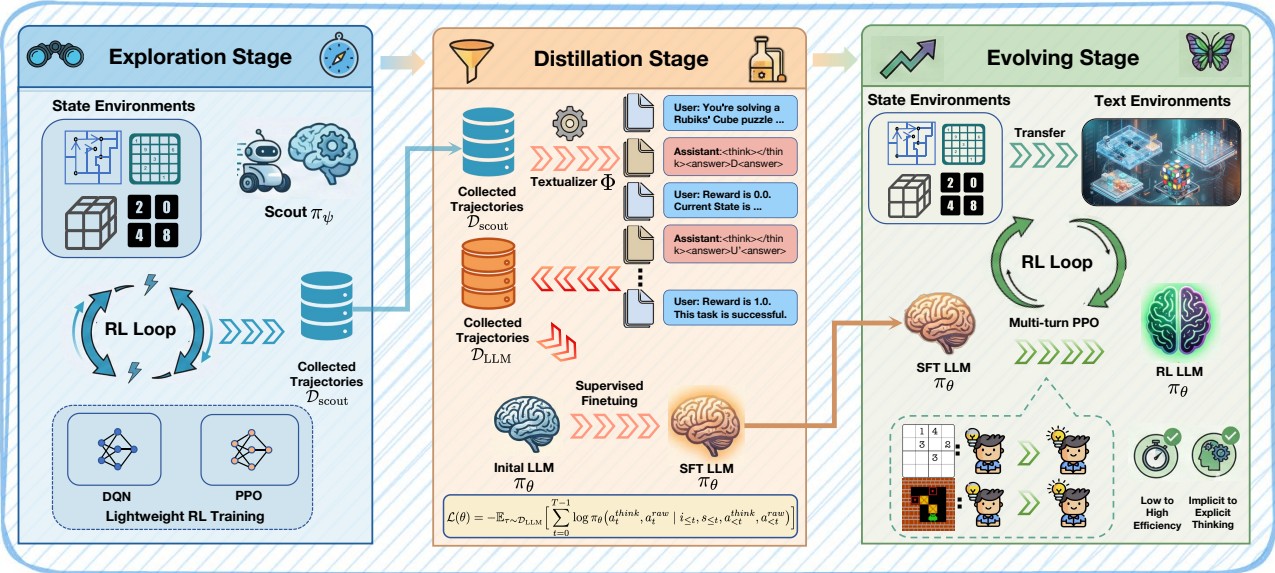

*Figure 2.* **Overview of the SCOUT framework**. The pipeline consists of three stages: (1) Exploration Stage: Lightweight scouts efficiently capture environmental dynamics to generate expert trajectories; (2) Distillation Stage: These trajectories are textualized to "warm-up" the LLM via supervised fine-tuning; (3) Evolving Stage: The LLM further refines its reasoning and decision making capabilities through multi-turn PPO.

tial tasks (Ghugare et al., 2025; Gu et al., 2023), symbolic tasks (Brockman et al., 2016) and complex long horizon tasks (Luo et al., 2025; Wang et al., 2025c), existing pretraining is far from sufficient. These tasks demonstrate that the real world is unbounded, involving "endless complexity" (Sutton, 2019). Therefore it is hard to fully simplify and cover all tasks during pretraining. LLMs' rich pretrained knowledge struggles in these scenarios because now they need to internalize the environmental dynamics from scratch rather than directly utilizing the pretrained world knowledge. SPA (Chen et al., 2025) attributes this performance gap to the fact that LLMs are significantly less familiar with symbolic state-based tasks (Brockman et al., 2016) compared to the language based state tasks like Webshop (Yao et al., preprint),ALFWorld (Shridhar et al., 2020). SPA separates the in-distribution and out-of-distribution tasks by the state perplexity against random guess. In this work, **we focus on these out-of-distribution tasks** that are often composed of various symbols or numbers, rather than natural language, and are more alien to language agents. **The newly introduced tasks in this work are also OOD tasks, and we show the higher state perplexity against random guess as evidence of OOD tasks as SPA (Chen et al., 2025) does in Table 8.** We mainly focus on these symbolic and spatial tasks, and call them "unseen tasks" in this work.

Beyond the pretraining stage, the inefficiency of LLM agents in mastering these new tasks also stems from two fundamental mismatches. **First**, there is a mismatch between the action space and the generation space. Generating a token requires a forward pass through billions of parameters of LLMs, resulting in low efficiency for both exploration and

exploitation. Furthermore, an LLM is exploring a vast vocabulary space (typically exceeding 30,000 tokens), whereas many reasoning and symbolic tasks (Brockman et al., 2016) require only a discrete, low dimensional set of actions. Although settings like temperature and top-k can reduce the candidate tokens, forcing an LLM to search for optimal policies within such a high-dimensional semantic space is computationally wasteful and hinders efficient exploration. **Second**, depending solely on language priors limits scalability. The "Bitter Lesson" (Sutton, 2019) teaches us that leveraging computation (searching and learning) is far more effective than relying on predefined knowledge in the long run. Although LLMs are semantically rich, they struggle to grasp the specific dynamics of the physical world (Ghugare et al., 2025; Wang et al., 2024; Yamada et al., 2023) that cannot be fully encoded in text.

To bridge this gap, we propose **SCOUT** (**S**ub-**S**cale **C**ollaboration **O**n Unseen **T**asks), a novel agent framework that harmonizes the exploitation on world knowledge of LLM agents with the exploration efficiency of "scouts". As shown in Figure 2, our key insight is to decouple the heavy exploration phase of LLM agents from the exploitation phase. Specifically, **we employ lightweight neural networks (e.g., small MLPs or CNNs) to serve as "scouts"**. Their low parameter count and high inference speed enable them to evolve rapidly with classic Reinforcement Learning (RL) algorithms (e.g., DQN, PPO) to master the environmental dynamics and generate high quality expert trajectories. These trajectories then serve as a warm-up for the LLM. This process effectively distills the specific task dynamics captured by the scouts into the LLMs, activating the LLMs'

internal relevant world knowledge with the specific unseen task. We further conduct multi-turn RL on the LLMs to align them with new tasks. This allows the LLMs to skip the heavy and inefficient exploration phase and focus on the exploitation of newly learned knowledge.

We test our methods on several symbolic and dynamic worlds such as FrozenLake, Sokoban and Sudoku. We further investigate the long horizon and spatial ability, and respectively introduce: 1) 2048, a grid game which needs above 800 turns to reach the 2048 tile; 2) Rubiks' Cube, a game of restoring a scrambled Rubiks' Cube. Empirical results demonstrate SCOUT significantly outperforms baselines. SCOUT enables a Qwen2.5-3B-Instruct model to achieve an average score of 0.86, significantly outperforming the tested baselines and proprietary models, within which Gemini-2.5-Pro achieves the highest 0.60 score. To summarize, we propose a novel framework, SCOUT. By leveraging small neural networks as scouts for rapid exploration and trajectory generation, we alleviate the exploration efficiency bottleneck inherent in pure LLM agents and activate the learned world knowledge with multi-turn RL. We demonstrate that SCOUT effectively activates the LLM's potential for unseen OOD tasks, allowing it to model the new environment efficiently and effectively.

## 2. Sub-Scale Collaboration On Unseen Tasks

In this section, we introduce the agentic framework SCOUT (Sub-Scale Collaboration On Unseen Tasks) from four aspects. First, we explain the Preliminaries. Then we introduce the Exploration Stage where the small neural network scouts self-evolve in the agentic task environments. We further explain the Distillation Stage that teaches the LLMs unseen task dynamics with expert trajectories. Finally, we introduce the Evolving Stage, which activates the learned knowledge in LLMs with multi-turn RL on the unseen tasks.

### 2.1. Preliminaries

We formulate the unseen tasks (e.g., symbolic tasks) as a Markov Decision Process (MDP) for the LLMs, and as a different MDP for the small neural network scouts.

**LLMs MDP** For LLMs, the environment is defined as a tuple $\mathcal{M}_{\text{LLM}} = \langle \mathcal{S}, \mathcal{I}, \mathcal{A}, \mathcal{P}, \mathcal{R} \rangle$. Here, $s_t \in \mathcal{S}$ represents symbolic states (e.g., grid matrices) at timestep $t$, and $i_t \in \mathcal{I}$ represents the language augmentations (e.g., task descriptions, transition rules). The LLM observes the full context $(i_t, s_t)$. $\mathcal{A}$ is the action space, $\mathcal{P}(s_{t+1}|s_t, a_t)$ denotes the transition dynamics, and $\mathcal{R}(s_t, a_t) \to \mathbb{R}$ provides the scalar reward signal. $\tau_{\text{LLM}} = \{i_0, s_0, a_0^{think}, a_0^{raw}, r_0, ..., i_T, s_T\}$ denotes the LLM interaction history.

**Scouts MDP** In contrast, for the scouts, we model the task as an intrinsic symbolic MDP defined by $\mathcal{M}_{\text{scout}} =$

$\langle \mathcal{S}, \mathcal{A}, \mathcal{P}, \mathcal{R} \rangle$. The scouts' observation consists of the symbolic state $s_t$. Unlike the LLM which could be hinted by language $i$ to infer environmental rules (e.g., "slippery ice"), the scout implicitly learns the underlying transition dynamics $\mathcal{P}$ directly through extensive trial-and-error. Consequently, the symbolic state serves as a sufficient statistic for the physical environment, allowing the scouts to master the dynamics without linguistic descriptors. The interaction history of the scouts is $\tau_{\text{scout}} = \{s_0, a_0, r_0, ..., s_T\}$.

**State-Text Mapping** To bridge the modality gap between $\mathcal{M}_{\text{scout}}$ and $\mathcal{M}_{\text{LLM}}$, we define a trajectory transformation function $\mathcal{T}$ that automatically converts scout experiences into multi-turn dialogue formats. Instead of requiring complex manual rule design, this function leverages the inherent interfaces of the environment to deterministically translate the symbolic trajectories $\tau_{\text{scout}}$ into their corresponding language dialogue $\tau_{\text{LLM}}$ by a Textualizer $\Phi$ without manual engineering, where thought content is set to blank. More details about the transformation are provided in Appendix A and Table 11. The detailed notations are listed in Table 7.

### 2.2. Exploration Stage

In this Exploration Stage, our primary objective is to bypass the inefficient exploration capability of Large Language Models by delegating the task of learning environmental dynamics to a lightweight proxy agent, denoted as the "scout". As formulated in the preliminaries, the scout operates within the reduced environment $\mathcal{M}_{\text{scout}}$, observing only the symbolic state $s_t$ without language augmentations. We parameterize the scout agent with learnable parameters $\psi$ using a lightweight neural network (e.g., an MLP or a small CNN), which is significantly smaller than the LLM $\pi_\theta$.

Given that the action spaces of the targeted tasks are discrete, we employ standard Reinforcement Learning algorithms, specifically DQN (Mnih et al., 2015) and PPO (Schulman et al., 2017) to train the scout. Our general goal is to maximize the expected cumulative reward:

$$J(\psi) = \mathbb{E}_{\tau \sim \pi} \left[ \sum_{t=0}^{T} \gamma^t r_t \right] \tag{1}$$

To achieve this, we adopt distinct optimization objectives depending on the algorithm employed. For **DQN**, where the policy is implicitly derived from value estimates, we approximate the optimal action-value function $Q_\psi$ by minimizing the temporal difference (TD) error against a target network:

$$\mathcal{L}_{\text{DQN}}(\psi) = \mathbb{E}_{(s_t, a_t, r_t, s_{t+1}) \sim \mathcal{B}} \Big[ \Big( r_t + \gamma \max_{a'_{t+1}}$$
$$Q_{\psi^-}(s_{t+1}, a'_{t+1}) - Q_\psi(s_t, a_t) \Big)^2 \Big] \tag{2}$$

where $\mathcal{B}$ denotes the replay buffer, $Q_\psi$ is the Q-network

parameterized by $\psi$, and $\psi^-$ represents the parameters of the frozen target network.

Conversely, when utilizing **PPO**, $\psi$ directly parameterizes the stochastic policy $\pi_\psi$. We maximize a clipped surrogate objective to ensure monotonic improvement without dangerously large policy updates:

$$\mathcal{L}_{\text{PPO}}(\psi) = \mathbb{E}_t \left[ \min \left( \rho_t(\psi) A_t, \text{clip}(\rho_t(\psi), 1 - \epsilon, 1 + \epsilon) A_t \right) \right] \tag{3}$$

where $\rho_t(\psi) = \frac{\pi_\psi(a_t|s_t)}{\pi_{\psi_{\text{old}}}(a_t|s_t)}$ is the probability ratio, $A_t$ is the estimated advantage, and $\epsilon$ controls the clipping range.

Due to the low dimensionality of the scout's parameter space and the absence of complex token generation overhead, the scout can interact with the environment at a frequency orders of magnitude higher than that of $\pi_\theta$. This high throughput interaction allows the scout to rapidly balance the exploration and exploitation trade-off, effectively mapping the transition dynamics and identifying high reward regions in the state space. After convergence, we utilize the better scout policy $\pi_\psi^*$ between DQN and PPO to generate a dataset of expert trajectories $\mathcal{D}_{\text{scout}} = \{\tau_1, \tau_2, \ldots, \tau_N\}$ on each task.

## 2.3. Distillation Stage

The second stage focuses on bridging the modality gap between the symbolic mastery of the scout and the linguistic reasoning of the LLM. The raw trajectories in $\mathcal{D}_{\text{scout}}$ lack the language context required by the LLM's input space. Therefore, we introduce a trajectory transformation function $\mathcal{T}$ defined in Preliminaries.

Formally, we define this trajectory transformation function $\mathcal{T}$ that converts the numerical scout trajectories into multi-turn dialogue formats. For each trajectory $\tau_{\text{scout}} = (s_0, a_0, r_0, s_1, a_1, \ldots, s_T)$ collected by the scout, we apply the Textualizer $\Phi$ to each item to reconstruct the linguistic context. The transformed trajectory $\tau_{\text{LLM}}$ is constructed as a sequence of dialogue turns:

$$\begin{aligned} \tau_{\text{LLM}} &= \mathcal{T}(\tau_{\text{scout}}) \\ &= \{ \underbrace{\Phi(s_0)}_{\text{User}}, \underbrace{\Phi(a_0)}_{\text{Asst}}, \underbrace{\Phi(r_0)}_{\text{User}}, \underbrace{\Phi(s_1)}_{\text{User}}, \ldots, \underbrace{\Phi(s_T)}_{\text{User}} \} \\ &= \{ i_0, s_0, a_0^{think}, a_0^{raw}, r_0, \ldots, i_T, s_T \} \end{aligned} \tag{4}$$

This results in an augmented dataset $\mathcal{D}_{\text{LLM}} = \{\mathcal{T}(\tau) \mid \tau \in \mathcal{D}_{\text{scout}}\}$, where symbolic state dynamics are explicitly grounded in language descriptions. Notably, we leave the $a^{think}$ blank: <think> </think>, as the trajectories do not include thinking content. We further analyze the emerging thinking content within the Evolving Stage in Section 4.6.

We then employ Supervised Fine-Tuning (SFT) to warm-up the LLM policy $\pi_\theta$ with $\mathcal{D}_{\text{LLM}}$. The optimization objective is to minimize the negative log-likelihood of the actions given the language-augmented context:

$$\begin{aligned} \mathcal{L}(\theta) = -\mathbb{E}_{\tau \sim \mathcal{D}_{\text{LLM}}} \Big[ \sum_{t=0}^{T-1} \log \pi_\theta \big( a_t^{think}, a_t^{raw} \mid \\ i_{\leq t}, s_{\leq t}, a_{<t}^{think}, a_{<t}^{raw} \big) \Big] \end{aligned} \tag{5}$$

This distillation process serves a critical purpose: it internalizes the "physics" of the unseen task into the LLM. Unlike standard pretraining where the model learns general world knowledge, this stage forces the LLM to align its internal representations with the specific, often counter intuitive dynamics of the unseen environment (e.g., the slipping mechanism in FrozenLake or the spatial permutations in Rubik's Cube). By cloning the behavior of the scout, $\pi_\theta$ effectively skips the prohibitively expensive initial exploration phase, starting its own learning curve from a distinct point of competence rather than random initialization.

## 2.4. Evolving Stage

In the final stage, we unleash the full potential of the LLM by transitioning from imitation to self-evolution. While the Distillation Stage equips the LLM with basic environmental dynamics and rule adherence, the policy $\pi_\theta$ is initially constrained by the upper bound of the scout's limited capacity and the supervised nature of the loss. To transcend these limitations, we conduct multi-turn Reinforcement Learning directly on the warm-up $\pi_\theta$ within the fully interactive environment $\mathcal{M}_{\text{LLM}}$.

**Trajectory-Level Optimization** Standard RLHF methods (Rafailov et al., 2023; Ouyang et al., 2022) typically treat alignment as a single-turn optimization. The objective is to maximize the expected reward of a single full response $y$ given a prompt $x$, constrained by a KL-divergence term to prevent deviation from the reference policy $\pi_{\text{ref}}$:

$$J_{\text{resp}}(\theta) = \mathop{\mathbb{E}}_{\substack{x \sim \mathcal{D} \\ y \sim \pi_\theta(\cdot|x)}} \left[ R(x, y) - \beta D_{\text{KL}}(\pi_\theta(\cdot|x) \parallel \pi_{\text{ref}}(\cdot|x)) \right] \tag{6}$$

However, this formulation overlooks the temporal dependencies in agentic tasks, where current actions $a_t$ (comprising both the thought process $a_t^{\text{think}}$ and execution $a_t^{\text{raw}}$) determine future states $s_{t+1}$ and the ultimate success of the episode. To address this, we employ **trajectory-level optimization** via multi-turn PPO (Wang et al., 2025d), aiming to maximize the expected cumulative return over the entire interaction

history $\tau$:

$$J_{\text{traj}}(\theta) = \mathbb{E}_{\tau \sim \pi_\theta}\left[\sum_{t=0}^{T}\left(\gamma^t r_t \right.\right.$$
$$\left.\left. - \beta D_{\text{KL}}(\pi_\theta(\cdot|h_t) \| \pi_{\text{ref}}(\cdot|h_t))\right)\right] \quad (7)$$

where $h_t = (i_t, s_t, \tau_{<t})$ denotes the full context defined in the preliminaries. Unlike in the Distillation Stage where the reasoning process is set to blank, in this stage we encourage the model to generate meaningful $\langle\text{think}\rangle$ blocks that serve as planning steps to maximize long term rewards.

**Activation and Refinement of Capabilities** This stage acts as a catalyst, interacting with the distilled knowledge in two distinct ways: refinement and activation. As shown in Figure 1, in environments like FrozenLake and Rubiks' Cube, the Distillation Stage alone grants the LLM substantial proficiency, indicating that the scout has successfully transferred the core task dynamics. Here, the multi-turn RL serves to refine this solid foundation, closing the gap to this almost perfect performance. Conversely, in tasks like Sudoku, the distilled policy initially appears limited, achieving a success rate of only 0.29. However, this seemingly low score masks a critical achievement: the LLM has internalized the valid rules but lacks strategic foresight. The subsequent RL rapidly "activates" this latent capability, boosting the score to 0.97. This dual effect demonstrates the versatility of SCOUT: whether the scout provides a near complete solution or a latent structural understanding, the subsequent RL effectively evolves this foundation into a strong policy.

## 3. Experiment Setup

**Models** For the trained models, we mainly use Qwen-2.5 series models (Yang et al., 2025) as our backbone, which aligns with our baselines (Wang et al., 2025d; Chen et al., 2025). We compare our method with existing baselines: RAGEN (Wang et al., 2025d), State Estimation RL (Chen et al., 2025), SPA (Chen et al., 2025), and some proprietary models like GPT-4o-mini (Hurst et al., 2024), DeepSeek-V3 (Liu et al., 2024), GPT-OSS-120B (Agarwal et al., 2025), GPT-5-nano (OpenAI, 2025), Gemini-2.5-Pro (Comanici et al., 2025) with reasoning activated system prompt as described in Appendix F.

**Tasks** We introduce 6 tasks with different settings and difficulty. **As illustrated in Introduction, we mainly focus on unseen tasks: tasks with higher state perplexity than random guess. We validate this in Appendix, Table 8.** We follow RAGEN (Wang et al., 2025d) to include Bandit, Frozen-Lake, Sokoban, Sudoku, and extend the environments to a long-horizon symbolic task: 2048 and a symbolic spatial task: Rubiks' Cube. Notably, for FrozenLake, Sokoban, Rubiks' Cube, they have different difficulties. The ground

of the FrozenLake could transfer from static to slippery, meaning that the action made by the agents could transfer from deterministic action to random action that "slips" on the ground. Increasing the box number of Sokoban could increase the difficulty of the Sokoban. Rubiks' Cube is a game that recovery a 2x2 Cube with 6 surface. The rotation number means the times that rotate the Cube from its intact state. Increasing rotation numbers make it harder to recovery the cube. The agents also need spatial imagination to accurately image the next spatial state. More details about each env's setting and difficulty are shown in Appendix D.1.

**Training Settings** For the multi-turn PPO in Evolving Stage, we conduct our experiments on RAGEN's codebase (Wang et al., 2025d), and follow their default setting to train the models for 200 steps. For the Supervised Fine-tuning(SFT) process in Distillation Stage, we employ LLaMA-Factory's codebase (Zheng et al., 2024). For the training of the scouts, we include cleanrl (Huang et al., 2022) as reference.

**Evaluation** We use the default codebase in RAGEN to eval the trained LLMs and the Proprietary Models by api to ensure fair comparison. The RAGEN codebase provides both the evaluation for the local models and api models. More details are provided in Appendix D.

## 4. Experimental Results and Findings

### 4.1. Main Experimental Results

We conduct detailed experiments on the 6 unseen tasks with different levels of difficulty: Whether the FrozenLake is slippery or not, the number of the boxes in Sokoban, the rotation times of the Rubiks' Cube. The details of each environment are introduced in Appendix D.1.

Table 1 shows the main experiments results. SCOUT significantly surpasses the baselines across various model sizes and types. With SCOUT, Qwen2.5-3B-It even beats several proprietary models like DeepSeek-V3, GPT-4o-mini, Gemini-2.5-Pro with an average score of 0.86. Increasing the model size from 0.5B to 3B consistently improves the performance from 0.81 to 0.86. On a different model type: LLaMA3.1, SCOUT also performs well that achieves a 0.83 score. Compared to the scouts, relying solely on the Distillation Stage is far from enough. Although the language agents learn the format and knowledge from the scout trajectories that achieve a score of above 0.6, even better than Multi-turn PPO, they still lag behind the performance of the scouts. Therefore, further RL on the SFT checkpoint is crucial that yields a performance gain of nearly 0.2. We could also observe the different task dynamics from this table. In tasks like Rubiks' Cube, the LLMs could already perform well with only SFT that achieves a nearly 90% win rate. However, on tasks like Sudoku, the learned dynamics from the expert trajectories need further RL to be activated.

*Table 1.* Main results on 6 unseen tasks. The 2048 score is normalized as (*Max-N*)/2048 to align with the [0,1] scale of other tasks. The scores represent the success rate (pass@1).

| Model/Task | Bandit | 2048 | | FrozenLake | | Sokoban | | Rubiks' Cube | | | Sudoku | Average |
|---|---|---|---|---|---|---|---|---|---|---|---|---|
| | | *Max-N* | *Return* | *Static* | *Slippery* | *Box1* | *Box2* | *Rotation1* | *Rotation2* | *Rotation3* | | |
| Small Neural Networks as Scouts | | | | | | | | | | | | |
| Scout-DQN | **0.93**$_{\pm0.02}$ | 1024 | **4319.40**$_{\pm251.38}$ | **0.90**$_{\pm0.05}$ | 0.80$_{\pm0.05}$ | 0.98$_{\pm0.01}$ | 0.47$_{\pm0.03}$ | **1.00**$_{\pm0.00}$ | **0.96**$_{\pm0.01}$ | **0.91**$_{\pm0.01}$ | 0.80$_{\pm0.04}$ | 0.83 |
| Scout-PPO | 0.79$_{\pm0.01}$ | 512 | 3677.64$_{\pm79.70}$ | **0.90**$_{\pm0.02}$ | **0.85**$_{\pm0.01}$ | **0.99**$_{\pm0.01}$ | **0.50**$_{\pm0.02}$ | **1.00**$_{\pm0.00}$ | 0.94$_{\pm0.01}$ | 0.81$_{\pm0.02}$ | **0.85**$_{\pm0.02}$ | 0.79 |
| SCOUT vs Baselines | | | | | | | | | | | | |
| Qwen2.5-0.5B-It | 0.39 | 128 | 47.11 | 0.17 | 0.14 | 0.04 | 0.00 | 0.14 | 0.08 | 0.03 | 0.00 | 0.11 |
| - Multi-turn PPO | 0.62 | 256 | 1091.57 | 0.39 | 0.24 | 0.15 | 0.06 | 0.45 | 0.22 | 0.11 | 0.00 | 0.24 |
| - State Estimation RL | 0.54 | 256 | 1193.56 | 0.27 | 0.24 | 0.20 | 0.06 | 0.31 | 0.23 | 0.10 | 0.05 | 0.21 |
| - SPA | 0.30 | 128 | 309.60 | 0.55 | 0.47 | 0.37 | 0.07 | 0.31 | 0.21 | 0.12 | 0.18 | 0.26 |
| - Exploration & Distillation Stage | 0.60 | **1024** | 5203.39 | 0.89 | 0.46 | 0.58 | 0.52 | 0.94 | 0.95 | 0.80 | 0.63 | 0.69 $_{\uparrow+0.58}$ |
| + Evolving Stage | **0.74** | **1024** | **5452.16** | **0.93** | **0.88** | **0.98** | **0.53** | **1.00** | **0.96** | **0.84** | **0.80** | **0.81** $_{\uparrow+0.70}$ |
| Qwen2.5-1.5B-It | 0.63 | 256 | 248.84 | 0.16 | 0.20 | 0.11 | 0.00 | 0.11 | 0.05 | 0.04 | 0.00 | 0.14 |
| - Multi-turn PPO | 0.72 | 256 | 1649.36 | 0.66 | 0.38 | 0.25 | 0.06 | 0.67 | 0.26 | 0.11 | 0.18 | 0.34 |
| - State Estimation RL | 0.71 | 512 | 2503.17 | 0.30 | 0.28 | 0.53 | 0.08 | 0.40 | 0.23 | 0.14 | 0.39 | 0.33 |
| - SPA | 0.23 | 512 | 2332.41 | 0.85 | 0.71 | 0.60 | 0.09 | 0.34 | 0.22 | 0.13 | 0.60 | 0.40 |
| - Exploration & Distillation Stage | **0.95** | **1024** | 4679.11 | 0.57 | 0.85 | 0.89 | 0.08 | 0.99 | 0.98 | 0.81 | 0.45 | 0.71 $_{\uparrow+0.57}$ |
| + Evolving Stage | **0.95** | **1024** | **5585.95** | **0.95** | **0.90** | **0.97** | **0.54** | **1.00** | **0.99** | **0.84** | **0.90** | **0.85** $_{\uparrow+0.71}$ |
| Qwen2.5-3B-It | 0.77 | 256 | 556.77 | 0.24 | 0.33 | 0.13 | 0.02 | 0.14 | 0.04 | 0.04 | 0.00 | 0.18 |
| - Multi-turn PPO | 0.87 | 256 | 1571.52 | 0.87 | 0.74 | 0.37 | 0.06 | 0.34 | 0.25 | 0.14 | 0.06 | 0.38 |
| - State Estimation RL | 0.63 | 256 | 1490.47 | 0.31 | 0.25 | 0.26 | 0.08 | 0.48 | 0.26 | 0.14 | 0.24 | 0.28 |
| - SPA | 0.23 | 512 | 1493.60 | 0.84 | 0.41 | 0.50 | 0.07 | 0.36 | 0.22 | 0.11 | 0.70 | 0.37 |
| - Exploration & Distillation Stage | 0.73 | **1024** | 5479.43 | 0.91 | 0.86 | 0.93 | 0.45 | 0.84 | 0.94 | 0.89 | 0.29 | 0.73 $_{\uparrow+0.55}$ |
| + Evolving Stage | **0.93** | **1024** | **5577.14** | **0.96** | **0.90** | **0.97** | **0.55** | **1.00** | **0.96** | **0.90** | **0.97** | **0.86** $_{\uparrow+0.68}$ |
| Qwen2.5-7B-It | 0.76 | 256 | 1103.05 | 0.53 | 0.46 | 0.35 | 0.02 | 0.30 | 0.06 | 0.10 | 0.08 | 0.28 |
| - Multi-turn PPO | 0.93 | 256 | 2308.76 | 0.89 | 0.82 | 0.43 | 0.04 | 0.49 | 0.24 | 0.11 | 0.74 | 0.48 |
| - Exploration & Distillation Stage | **1.00** | **1024** | 4834.59 | 0.91 | 0.87 | 0.71 | 0.54 | 1.00 | 0.92 | 0.90 | 0.41 | 0.78 $_{\uparrow+0.50}$ |
| + Evolving Stage | **1.00** | **1024** | **5849.62** | **0.96** | **0.91** | **0.95** | **0.57** | **1.00** | **1.00** | **0.93** | **0.98** | **0.88** $_{\uparrow+0.60}$ |
| LLaMA3.1-1B-It | 0.43 | 256 | 741.54 | 0.16 | 0.19 | 0.08 | 0.01 | 0.01 | 0.00 | 0.00 | 0.00 | 0.10 |
| - Multi-turn PPO | 0.63 | 256 | 2173.45 | 0.37 | 0.38 | 0.26 | 0.07 | 0.38 | 0.02 | 0.09 | 0.03 | 0.24 |
| - Exploration & Distillation Stage | 0.80 | 128 | 39.82 | 0.88 | 0.84 | 0.11 | 0.32 | 1.00 | 0.96 | 0.84 | 0.44 | 0.63 $_{\uparrow+0.53}$ |
| + Evolving Stage | **0.81** | **1024** | **3697.29** | **0.91** | **0.88** | **0.95** | **0.51** | **1.00** | **1.00** | **0.85** | **0.92** | **0.83** $_{\uparrow+0.73}$ |
| Proprietary Models | | | | | | | | | | | | |
| GPT-4o-mini | 0.73 | 256 | 1507 | 0.94 | 0.84 | 0.34 | 0.10 | 0.02 | 0.00 | 0.00 | 0.71 | 0.38 |
| DeepSeek-V3 | 0.81 | 256 | 3864 | 0.93 | 0.85 | 0.45 | 0.22 | 0.06 | 0.00 | 0.00 | 0.93 | 0.44 |
| GPT-OSS-120B | 0.66 | 256 | 3320 | 0.95 | 0.88 | 0.95 | 0.71 | 0.19 | 0.19 | 0.00 | 1.00 | 0.57 |
| GPT-5-nano | 0.71 | 256 | 1690 | 0.84 | 0.66 | 0.96 | 0.56 | 0.00 | 0.00 | 0.00 | 1.00 | 0.49 |
| Gemini-2.5-Pro | 0.69 | 256 | 2436 | 1.00 | 0.88 | 0.97 | 0.59 | 0.31 | 0.28 | 0.16 | 0.97 | 0.60 |

## 4.2. Surpass the Subagent in the Unseen World

**Scout Comparison** In Table 1, we compare two differ-ent scouts' abilities. We initialize small neural networks (MLPs for Bandit, 2048, FrozenLake, Rubiks' Cube, Su-doku; CNNs for Sokoban) and compare the performance of Deep Q-Network (DQN) with that of Proximal Policy Opti-mization (PPO). The detailed small neural network scouts' architectures are shown in Appendix G.1.

As evidenced by the results, Scout-DQN generally outper-forms Scout-PPO across some of the evaluated metrics. The Scout-DQN achieves superior or equal best performance in

4 out of the 10 detailed tasks, and often by a significant mar-gin (e.g., achieving a value of 1024 in the second column compared to PPO's 512), equal performance in 2 tasks, and fall behind in the other 4 tasks. While Scout-PPO shows competitive results in certain tasks (e.g., scoring 0.85 and 0.85 in the FrozenLake Slippery and Sudoku, respectively, against DQN's 0.80 and 0.80), it does not match the general performance of DQN. This empirical advantage of DQN is likely due to the discrete nature of the action spaces in the tested environments, where off-policy value-based methods often demonstrate higher sample efficiency than on-policy methods like PPO (Schulman et al., 2017; Mnih et al., 2015).

**Exploration Efficiency** It is notable that, after conducting multi-turn PPO on the SCOUT Distillation checkpoint, the learned but not yet demonstrated capability is effectively activated. With SCOUT, the Qwen2.5-3B-It and Qwen2.5-1.5B-It even surpass the average performance of Scout-DQN and Scout-PPO. This result validates our core hypothesis: the bottleneck of language agents in unseen tasks lies less in the reasoning capacity, but more in the efficiency of initial exploration. By leveraging the Scout to handle the heavy burden of trial-and-error, the LLM effectively bypasses the computationally expensive phase of learning environmental dynamics from scratch. Instead, it directly focuses on exploitation and high level reasoning, seamlessly integrating the distilled "physics" of the task with its intrinsic semantic capabilities. Consequently, the agent not only masters the symbolic mechanics faster but also transcends the limits of its teacher (the Scout), demonstrating that the "Sub-Scale Collaboration" can unlock latent potential while preserving the versatility of the language model. This stark contrast is quantified by the difference in parameter size and memory footprint. As shown in Table 2, Scouts utilize approximately $1.0 \times 10^{-5}$ billion parameters which are nearly $10^5$ times smaller than the LLMs, allowing them to operate faster. This lightweight nature effectively decouples the exploration phase from expensive GPU resources, transforming the high-cost, low-efficiency trial-and-error process of LLMs into a computationally inexpensive task. Thus, SCOUT achieves superior exploration coverage with minimal energy consumption.

*Table 2.* **Resource Efficiency Comparison.** Scouts operate primarily on CPUs, demonstrating significant reductions in hardware dependency and resource consumption compared to LLMs.

| Metric | LLMs | Small Scouts | Efficiency Gain |
|---|---|---|---|
| **Training Device** | High-end GPU | Commodity CPU | *GPU Independent* |
| **Parameter Size** | 0.5B – 3B | $\sim 1.0 \times 10^{-5}$ B | $\sim 10^5 \times$ **Smaller** |
| **Memory Footprint** | $> 40$ GB (VRAM) | $< 1$ GB (RAM) | $\sim 10^2 \times$ **Lower** |

**GPU Cost Analysis** To further quantify the computational efficiency, we perform a detailed cost analysis on the challenging Rubiks' Cube Rotation3 task using Qwen2.5-3B-Instruct, as shown in Table 3. The Direct PPO baseline incurs a substantial computational overhead, consuming **24.0 GPU hours** (on an $8\times$H100 node) to complete 200 training steps. This inefficiency arises because the heavy LLM is forced to perform the entire trial-and-error exploration process on expensive GPU hardware. In contrast, SCOUT strategically optimizes resource allocation. By delegating the initial exploration to the lightweight Scout, we incur nearly zero GPU cost during the most uncertain phase of learning. The GPU resources are subsequently utilized only for efficient knowledge transfer (SFT) and activation (PPO). Consequently, SCOUT achieves the same training milestone with a total cost of only **9.6 GPU hours**, representing a dramatic $\sim$**60% reduction** in computational expense. The

significant speedup (24.0h vs 9.6h) stems from the context efficiency. Direct PPO involves long exploration trajectories that fill the thought content $a^{think}$. In contrast, SCOUT's Evolving Stage starts from high quality, concise expert paths (with blank thoughts $a^{think}$), which involves far fewer token counts and accelerating the optimization process. This result confirms that SCOUT provides an economically viable path for scaling RL to complex, long horizon tasks.

*Table 3.* **Training Cost Comparison on Rubiks' Cube Rotation3.** We report wall-clock time and GPU hours for 200 training steps. SCOUT significantly reduces total time and cost.

| Method | Stage | Device | Per Stage | | Total Cost | |
|---|---|---|---|---|---|---|
| | | | Time | GPU-h | Time | GPU-h |
| **Direct PPO** | RL Training | $8\times$H100 | 3.00 | 24.0 | 3.00 h | 24.0 |
| **SCOUT** | Exploration | CPU Only | 0.17 | 0.0 | | |
| | Distillation | $8\times$H100 | 0.20 | 1.6 | **1.37 h** | **9.6** |
| | Evolving | $8\times$H100 | 1.00 | 8.0 | | *(-60%)* |

Furthermore, to evaluate the robustness of our framework against noisy, low-quality data or scenarios where a high-quality expert scout is unavailable, we conduct an ablation study utilizing a *sub-optimal scout*. This proxy agent is constructed using an under-converged checkpoint extracted at merely 10% of the total training iterations, yielding a very limited success rate (only 0.34 in Rubik's Cube Rotation3 and 0.55 in Sudoku). As presented in Table 4, despite being bootstrapped by these low-quality and noisy trajectories during the Distillation Stage, the LLM agent via SCOUT drastically transcends its teacher, achieving scores of 0.60 and 0.98 respectively. This proves that the Exploration & Distillation Stage does not merely perform behavior cloning but successfully injects noisy world knowledge and unlocks better reasoning upon subsequent Evolving Stage.

*Table 4.* **Robustness Study with Sub-optimal Scout Initialization.** We bootstrap Qwen2.5-3B-It using a sub-optimal proxy agent to simulate scenarios where high-quality expert trajectories are unavailable. Despite low-quality initialization, SCOUT successfully learns the task dynamics and achieves high proficiency.

| Model / Task | Rubik's Cube Rotation3 | Sudoku |
|---|---|---|
| Sub-optimal Scout | 0.34 | 0.55 |
| Qwen2.5-3B-It | 0.04 | 0.00 |
| - with Direct RL | 0.14 | 0.06 |
| - Exploration & Distillation Stage | 0.27 | 0.42 |
| + Evolving Stage | **0.60** | **0.98** |

### 4.3. Enabling Multi-task Language Agents via Sequential RL with SCOUT

Previous sections have shown the great potential of SCOUT in those single tasks. However, it remains a question whether the SCOUT paradigm could extend to a multi-task setting. The critical question is whether the language agents will **collapse** and **fall into catastrophic forgetting** when con-

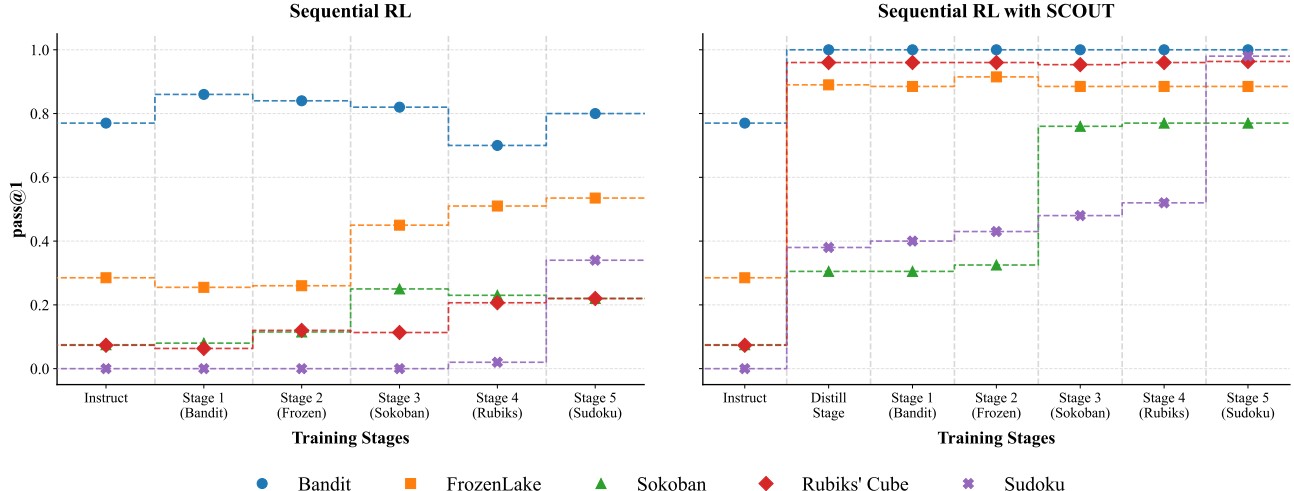

*Figure 3.* Comparison of task performance during sequential RL. While the Sequential RL (Left) exhibits some performance degradation on previously learned tasks, SCOUT (Right) successfully preserves historical task knowledge (e.g., Bandit, FrozenLake) while adapting to new environments (e.g., Sudoku), achieving a near optimal multi-task agent.

ducting multi-task training, or **maintain the trained task ability while extending their skills to new tasks?**

In this section, we conduct multi-task sequential RL on the language agents. We design a sequential curriculum order: Bandit → FrozenLake → Sokoban → Rubik's Cube → Sudoku. We evaluate two distinct settings as shown in Table 9: 1) Direct Sequential RL, where we directly apply PPO on the initial model sequentially; and 2) Sequential RL with SCOUT, where we first apply multi-task SFT using trajectories collected by scouts from all environments, followed by the same sequential PPO. We average the scores under different settings for the same task.

**The Role of Scout Initialization** Comparing the first two phases in Figure 3, we observe that the necessity of the Exploration and Distillation Stage is evident. Without the warm-up by scout trajectories, the Direct Sequential RL (left figure) struggles to explore effectively. The average score only marginally improves from 0.19 to 0.37 after five tasks of PPO. In the Bandit and Sokoban tasks, it even experienced performance fluctuations and declines. In contrast, the SCOUT paradigm (right figure) starts with a strong foundation (Multi-task SFT via Exploration and Distillation Stage) and further evolves to a multi-task expert with an average score of 0.91. This confirms that the lightweight Scouts effectively compress the dynamics of multiple unseen tasks into the LLM, providing a robust initialization for subsequent evolution.

**Plasticity and Stability Tradeoff** A major concern in multi-task learning is catastrophic forgetting, where learning a new task degrades performance on previously learned tasks. As observed in the right side of Figure 3, our approach demonstrates remarkable stability. For instance, after the

agent finishes the final training stage on Sudoku (Stage 5), it not only masters the new Sudoku task (from 0.38 after SFT to 0.98) but also retains high proficiency in the learned tasks. Specifically, the Bandit score remains stable at 1.0, and the FrozenLake scores (0.89) stay comparable to their initial post-SFT levels (0.89). Moreover, we observe positive transfer in complex tasks; for example, training on Sokoban and Rubik's Cube appears to aid the reasoning required for Sudoku, which improves significantly in performance. This suggests that the SCOUT framework allows the LLM to internalize a generalized "world model" rather than overfitting to isolated tasks, effectively mitigating catastrophic forgetting while continuously expanding its capability boundaries.

### 4.4. Generalization to Vision-based POMDP Environments

A common constraint for text-based symbolic agents is their heavy reliance on flawless, lossless serialization of environment states. To demonstrate that SCOUT is universally applicable and capable of overcoming structural observation boundaries, we scale our framework to a continuous, vision-based Partially Observable MDP (POMDP) setting using Qwen2.5-VL-3B-Instruct. In this configuration, instead of receiving clean discrete character matrix strings, the environment inputs are high-dimensional, continuous visual images (Visual-FrozenLake and Visual-Sokoban), where lightweight CNN-based vision scouts are used.

As summarized in Table 5, pure visual RL training from scratch on the VLM experiences exploration bottlenecks. In stark contrast, SCOUT bridges the multi-modal distribution gap effectively, providing significant performance leaps (boosting Visual-Sokoban from 0.57 to 0.95). This identifies SCOUT's strong generalization from text-based

modeling to Vision-based modeling.

*Table 5.* **Generalization to Vision-based POMDP Environments.** We evaluate SCOUT on continuous image inputs to verify its robustness against partial observability. We also report the out-of-distribution (OOD) cross-task transfer performance from Visual-FrozenLake to Visual-Sokoban.

| Model / Task | Visual-FrozenLake | Visual-Sokoban |
|---|---|---|
| Visual Scout | 0.94 | 0.90 |
| Qwen2.5-VL-3B-It | 0.10 | 0.14 |
|   - Direct RL | 0.70 | 0.57 |
|   - Exploration & Distillation Stage | 0.91 | 0.36 |
|     + Evolving Stage | **0.95** | **0.95** |

## 4.5. Generalization Across Task Difficulty

In this section, we test the generalization ability of SCOUT framework across the task difficulty. We select Rubiks' Cube as the test task as it naturally contains several different levels. We select Qwen2.5-3B-It as the backbone model. As shown in 6, models trained only on lower difficulty settings (such as Rotation 3) successfully generalize to harder instances (Rotation 4,5), this demonstrates extrapolation to unseen task complexity.

*Table 6.* **Generalization Across Task Difficulty on Rubik's Cube.** We report the success rate (pass@1) of models trained on lower complexity settings (shorter rotation depths) and evaluated zero-shot on harder instances (up to Rotation 5).

| Train / Test | Rotation 2 | Rotation 3 | Rotation 4 | Rotation 5 |
|---|---|---|---|---|
| Rotation 1 | 0.33 | 0.31 | 0.13 | 0.12 |
| Rotation 2 | — | 0.43 | 0.35 | 0.19 |
| Rotation 3 | — | — | **0.58** | **0.45** |

## 4.6. From Implicit Modeling to Explicit Modeling

At the Distillation Stage in Section 2.3, we leave the thinking content blank in $\mathcal{D}_{\text{LLM}}$. However, we explicitly require the language models to first output their thinking content, then follow their final answer in the multi-turn RL process in Evolving Stage. We find the RL finetuned language models fill in the blank between these thinking tags. The language models sometimes will directly output their intended action within the thought block before repeating it in the answer section. This is very obvious in tasks whose action space is short and simple, like FrozenLake, Rubiks' Cube. However, for tasks that need more language to answer, like Sudoku, the language models tend to output an analysis in the thinking part, then make their final decisions. In the following table, the RL trained language model successfully discovers the missing number in the three rows and three columns, and correctly outputs the answer.

## 5. Conclusion

In this paper, we identify the exploration inefficiency and dimension mismatch as key barriers for LLM agents in mastering unseen, non-linguistic tasks. To address these challenges, we propose SCOUT, a novel framework that harmonizes the rapid exploration of lightweight "scout" networks with the reasoning capabilities of LLMs. By decoupling exploration from exploitation, SCOUT efficiently distills environmental dynamics into the LLM, followed by further evolution via multi-turn RL. Empirical results across symbolic and spatial tasks, including long horizon challenges like 2048 and Rubiks' Cube, demonstrate that SCOUT significantly outperforms existing baselines. Further ablation studies on the generalization ability of SCOUT to POMDP environments, task difficult levels illustrate the strong potiential. Our work validates that "Sub-Scale Collaboration" is a promising path to bridge the gap between linguistic priors and the physical dynamics of the real world.

---

### Examples of Implicit to Explicit Modeling

```
<|im_start|>system
```
You are a helpful assistant. `<|im_end|>`
`<|im_start|>`user
You are solving a Sudoku puzzle. Fill in the grid so that every row, column, and 2x2 box contains the numbers 1-4 without repetition. Initial cells are shown in [brackets] and cannot be modified. Empty cells are shown as dots (.). Place numbers one at a time using the format, for example: <answer>place 1 at row 2 col 3</answer> or <answer>1,2,3</answer> The environment will provide feedback on valid/invalid moves and show conflicts.
Turn 1:
===============
SUDOKU PUZZLE
===============
[1] . | [3][2]
[2][3]| [1] .
—
[4] . | . .
[3][2]| [4] .
Legend: [N]=initial cell, .=empty
Steps: 0/20
You have 20 actions left. Always output: <think> [Your thoughts] </think> <answer> [your answer] </answer> with no extra text. Strictly follow this format. Max response length: 150 words (tokens).
`<|im_end|>`
`<|im_start|>`assistant
<think>Left col 3 in row 3 should contain 2</think><answer>place 2 at row 3 col 3</answer>`<|im_end|>`

## Acknowledgements

Dr Tao's research is supported by the NTU Research Support Resources Award Number #023791-00001 and NTU Start Up Grant.

## Impact Statement

This paper presents SCOUT, a framework that significantly improves the efficiency of LLM agents in unseen environments. By offloading the computationally expensive exploration phase to lightweight neural networks, our approach addresses the "computational wastefulness" of using large models for trial-and-error tasks. This has positive implications for Green AI, as it reduces the energy footprint required to train competent agents. Furthermore, by demonstrating that smaller models (e.g., 3B parameters) can outperform larger proprietary ones through effective collaboration, our work promotes the democratization of capable AI agents, allowing researchers with limited computational resources to achieve state-of-the-art results. We do not foresee immediate negative societal consequences, though the advancement of autonomous agents warrants standard ethical monitoring regarding their deployment in real-world automated systems.

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

# Appendix for

## *Language-based Trial and Error Falls Behind in the Era of Experience*

## Part I: Background

## Part II: Experiments Setup

## Part III: Prompts and Architectures

# A. Notation

In this section, we list the detailed notation that we adopt in the main text.

*Table 7.* Summary of important notations used in the SCOUT framework.

| Notation | Symbol | Description |
|---|---|---|
| | | Language Models |
| State | $s_t$ | Ground-truth environment state at turn $t$. |
| Think Tokens | $a_t^{think}$ | The think tokens at turn $t$. |
| Raw Action | $a_t^{raw}$ | The raw action tokens at $t$, not augmented with think tokens. |
| Reward | $r_t$ | Scalar reward at turn $t$, $r_t = R(s_t, a_t)$ where $R$ is the reward function. |
| Language Augmentation | $i_t$ | The augmented text at turn $t$. |
| Trajectory | $\tau$ | Rollout $\tau = (i_0, s_0, a_0^{think}, a_0^{raw}, r_0, \cdots, i_T, s_T)$. |
| Language Agent Policy | $\pi_\theta$ | Policy parameterized by a LLM with parameters $\theta$. |
| | | Scouts |
| State | $s_t$ | Ground-truth environment state at turn $t$. |
| Raw Action | $a_t$ | The raw action tokens at $t$, not augmented with think tokens. |
| Reward | $r_t$ | Scalar reward at turn $t$, $r_t = R(s_t, a_t)$ where $R$ is the reward function. |
| Trajectory | $\tau$ | Rollout $\tau = (s_0, o_0, a_0, r_0, \cdots, s_T)$, not augmented by $I$. |
| Scout Policy | $\pi_\psi$ | Policy parameterized by a neural network with parameters $\psi$. |
| Token Index | $i, j$ | $\bar{\tau}_i$: the $i$-th token. $\bar{\tau}_{i:j}$: tokens $i$–$j$. $\bar{\tau}_{<i}$: prefix up to $i-1$. $\bar{\tau}_{t,i}$: the $i$-th token of the $t$-th turn. |
| Trajectory Return | $R(\tau)$ | Sum of rewards over trajectory, $\sum_t R(s_t, a_t)$. |
| Advantage Estimate | $A$ | $A_i$: advantage for token $i$; $A_t^{\text{turn}}$: advantage for turn $t$; $A_{t,i}^{\text{token}}$: advantage for token $i$ in turn $t$. |
| Discount Factor | $\gamma$ | $\gamma \in [0, 1]$; $\gamma_{\text{token}}$: within-turn discount; $\gamma_{\text{turn}}$: across-turn discount. |

# B. Limitations

In this work, we validate the efficiency of SCOUT framework on different model sizes, from 0.5B to 3B and on different model types, from Qwen to LLaMA. Due to the resource limitation, we do not validate larger size models or other type models, which may perform better than existing models. We mainly conduct our experiments with the mostly used and stable multi-turn PPO in this work, while other RL algorithms like GRPO (Guo et al., 2025) may also be effective. In some tasks, we observe the performance degradation after several RL training steps, which aligns with the findings in RL community (Wang et al., 2025d; Xue et al., 2025). Further improvements on stabilizing multi-turn RL training is essential.

# C. Related Works

**LLM Agents and Environment Interaction.** Recent studies (Yao et al., 2022; Liu et al., 2025a; Zhou et al., 2025a; Luo et al., 2025) have explored the capacity of Large Language Models (LLMs) to master complex environments through multi-turn interaction. These benchmarks range from text-based scenarios like ALFWorld (Shridhar et al., 2020), WebShop (Yao et al., preprint), TauBench (Yao et al., 2024), and GAIA (Mialon et al., 2023) to symbolic reasoning tasks such as FrozenLake (Brockman et al., 2016). Existing research primarily aims to enhance performance by optimizing reinforcement learning (RL) algorithms (Wang et al., 2025d; Xue et al., 2025), incorporating memory-based architectures (Zhou et al., 2025b; Jin et al., 2025), filtering instruction-tuning datasets (Xue et al., 2025), or converting textual inputs into visual representations (Wang et al., 2025b). In contrast, our work focuses on decoupling the computationally expensive exploration phase from the reasoning phase by introducing lightweight "scout" networks.

**Deep RL and Exploration Efficiency.** Deep Reinforcement Learning (DRL) has achieved significant success in mastering environmental dynamics (Mnih et al., 2015; Schulman et al., 2017; Haarnoja et al., 2018), ranging from Atari games to robotic control (Gu et al., 2023; Brockman et al., 2016). A key advantage of classical DRL is its ability to optimize policies within compact state spaces, enabling high throughput interaction that captures the underlying dynamics of the environment. We leverage this efficiency by employing lightweight networks (e.g., MLPs or CNNs) as "scouts." With significantly fewer parameters than LLMs, these scouts can rapidly balance the exploration-exploitation trade-off to generate expert trajectories, effectively addressing the "cold start" problem for the subsequent language agent.

**Large-Small Model Collaboration.** Collaborative frameworks involving large and small models have also gained much

attention (Zeng et al., 2026; Wang et al., 2025a; Liu et al., 2025b; Huang et al., 2026; Yu et al., 2025). Typically, a larger model acts as a planner while a smaller language model handles execution or tool use. Unlike these approaches, SCOUT employs non-linguistic neural networks to address the exploration bottleneck. Crucially, these scouts operate without pretrained linguistic knowledge, learning environmental dynamics from scratch. This design separates physical rule acquisition from semantic reasoning, allowing the LLM to learn from grounded experience via distillation rather than relying on the small model for runtime inference.

# D. Experiments

## D.1. Models, Datasets, Tasks

**Models** We follow previous agentic methods (Wang et al., 2025b;d; Chen et al., 2025), we utilize models of varying sizes and types.

- We adopt the instruct version of Qwen2.5B series models (Yang et al., 2025) as the backbone of finetuning. The sizes vary from 0.5B to 1.5B, 3B. We complement LLaMA3.1-1B-It (Grattafiori et al., 2024) to validate the SCOUT on different model types.

- We adopt several strong property models as our baselines. For proprietary solutions, we employ GPT-4o-mini (Hurst et al., 2024) as a representative of cost-effective agents, DeepSeek-V3 (Liu et al., 2024) for its robust reasoning, Gemini-2.5-Pro (Comanici et al., 2025) for its superior general ability, and the newest OpenAI model GPT-5-nano (OpenAI, 2025). We also evaluate high-performing open-source models, specifically the powerful GPT-OSS-120B (Agarwal et al., 2025).

- We adopt MLPs for Bandit, 2048, FrozenLake, Rubiks' Cube, Sudoku and CNNs for Sokoban. These small neural networks are only about $1.0 \times 10^{-5}$ B that interact very fast with the environments.

**Datasets** In this work, the included datasets are $\mathcal{D}_{\text{scout}}$ and $\mathcal{D}_{\text{LLM}}$.

- For $\mathcal{D}_{\text{scout}}$, we collect 4k trajectories each task. We utilize the final checkpoint of the trained scout to collect these trajectories. The collected $\tau_{\text{scout}} = (s_0, a_0, r_0, s_1, a_1, \ldots, s_T)$, where the state is represented by one-shot vector.

- For $\mathcal{D}_{\text{LLM}}$, we utilize the predefined trajectory transformation function $\mathcal{T}$ to convert the dataset $\mathcal{D}_{\text{scout}}$ into the multi-turn dialogue dataset $\mathcal{D}_{\text{LLM}}$. The trajectories in $\mathcal{D}_{\text{LLM}}$ is $\tau_{\text{LLM}} = \{i_0, s_0, a_0^{think}, a_0^{raw}, r_0, \ldots, i_T, s_T\}$, where we leave the $a^{think}$ blank.

**Tasks** We follow SPA (Chen et al., 2025) and **focus on out of distribution tasks** that often compose of various symbols or numbers, rather than natural language, and are more unseen to language agents. We mainly focus on the symbolic and spatial tasks whose state perplexity are larger than random guess as shown in table 8, and we call these tasks as "unseen tasks" in this work. We introduce two new tasks: 2048 and Rubiks' Cube.

- **Bandit:** A fundamental reinforcement learning benchmark serving as a sanity check. The agent must interact with two arms, each associated with a specific reward probability distribution, to identify and select the optimal arm for maximizing cumulative returns.

- **FrozenLake:** A grid world navigation task where the agent must reach a goal while avoiding holes. We evaluate two difficulty settings: *Static* and *Slippery*. In the *Static* setting, transitions are deterministic. In the *Slippery* setting, the ground is simulated as frictionless ice, meaning the agent may move in a direction perpendicular to the chosen action with a certain probability (i.e., slipping). This tests the agent's robustness against stochastic environmental dynamics.

- **Sokoban:** A classic planning puzzle requiring the agent to push boxes to designated target locations without getting stuck. We control the difficulty by varying the **number of boxes** (e.g., *Box1*, *Box2*). Increasing the box number exponentially expands the state space and increases the likelihood of irreversible deadlocks, demanding complex multistep reasoning and path planning.

- **Sudoku:** A logic-based combinatorial number-placement puzzle. The agent must fill a $4 \times 4$ grid such that every row, column, and subgrid contains unique digits. This task serves as a testbed for pure symbolic constraint satisfaction and deductive reasoning.

- **2048:** A long-horizon symbolic task where the agent slides and merges numbered tiles on a grid to reach the target number 2048. Unlike other short horizon tasks, a successful game typically requires more than 800 turns. This environment challenges the agent's ability to plan strategically for long term sustainability and maintain grid tidiness over a long horizon.

- **Rubik's Cube:** A spatial intelligence and symbolic task that requires restoring a scrambled $2 \times 2$ cube to its original state. We define the difficulty based on the **"rotation number"** (or scramble depth), which represents the number of random rotations applied to an intact cube to generate the initial state (e.g., *Rotation1*, *Rotation2*, *Rotation3*). Higher rotation numbers increase the complexity of restoration, requiring the agent to possess strong spatial imagination to mentally simulate 3D state transitions.

*Table 8.* **Quantifying Distribution Shift:** Average Perplexity (PPL) of state representations evaluated with Qwen2.5-Instruct-1.5B. Comparison between our symbolic tasks and standard language-based agent tasks. The significantly higher PPL than random guess in symbolic tasks (e.g., Sokoban, Frozen Lake) indicates that these environments are essentially **Out-of-Distribution (OOD)** and "unseen" to the LLM, contradicting the concern of data contamination. However, for language-base tasks like WebShop and ALFWorld, the PPLs are smaller than the random guess, indicating that they are much more in-distribution tasks.

| Task Environment | PPL (Perplexity) | Random Guess |
|---|---|---|
| ***Symbolic / Unseen Tasks*** | | |
| Sokoban | **163.90** | 7 |
| Frozen Lake | **187.10** | 6 |
| Rubiks' Cube | **24.38** | 6 |
| 2048 | **15.85** | 12 |
| Sudoku | **15.50** | 5 |
| ***Language / In-Distribution Tasks (Reference)*** | | |
| WebShop | 11.70 | Vocabulary Size |
| ALFWorld | 6.00 | Vocabulary Size |

### D.2. Experiment Settings

In this work, we utilize SFT and multi-turn PPO to train the models. This lead to several hype-parameters.

- We conduct SFT with LLaMA-Factory (Zheng et al., 2024). The training configuration includes a cutoff length of $4096$, a batch size of $64$, 3 training epochs, a cosine learning rate scheduler, and a warm-up ratio of $0.1$. For full finetuning, we set learning rate to $1e-5$. We conduct the training on an 8 H100 device.

- We conduct multi-turn PPO with RAGEN (Wang et al., 2025d). The training configuration includes an 0.28 clip ratio, an 0.25 rollout filter ration. We train all the checkpoint for 200 steps, keeping in line with RAGEN (Wang et al., 2025d). We set the max model len to 16384 to avoid the unexpected dialog cutoff. We request the agent to give one action per turn, and set the max turn to 25, except 2048, where we set the max turn to 1k. Also, we include a in-context sliding window in 2048 with 5 dialogue segments. This greatly reduces the pressure of the context, making it possible for the model to complete such a long horizon task. We conduct the multi-turn RL training on an 8 H100 devices.

## E. Extra Experimental Results

In this section, we give the detailed scout training curves on the 6 unseen tasks and their different settings as Figure 4 and Figure 5 show. We also include the detailed results on the sequential RL in Table 9. This sequential RL results correspond to the Figure 3 in the main context.

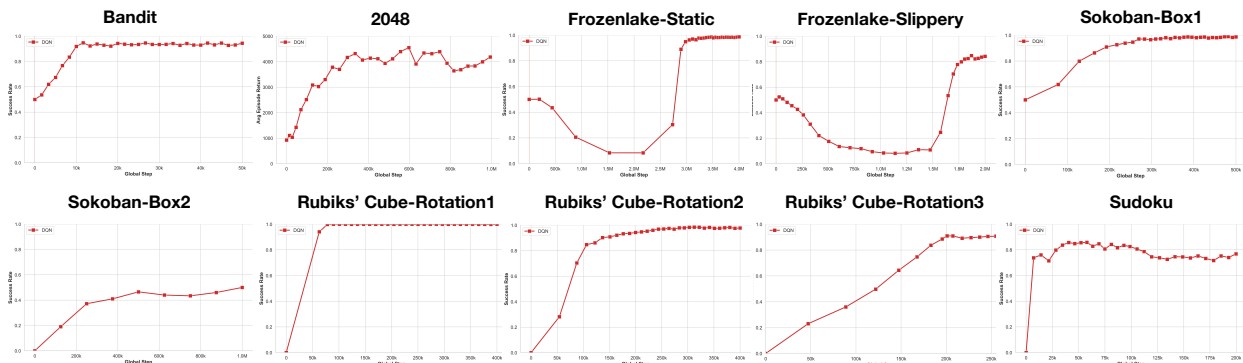

*Figure 4.* Scout-DQN detailed performance on 6 unseen tasks.

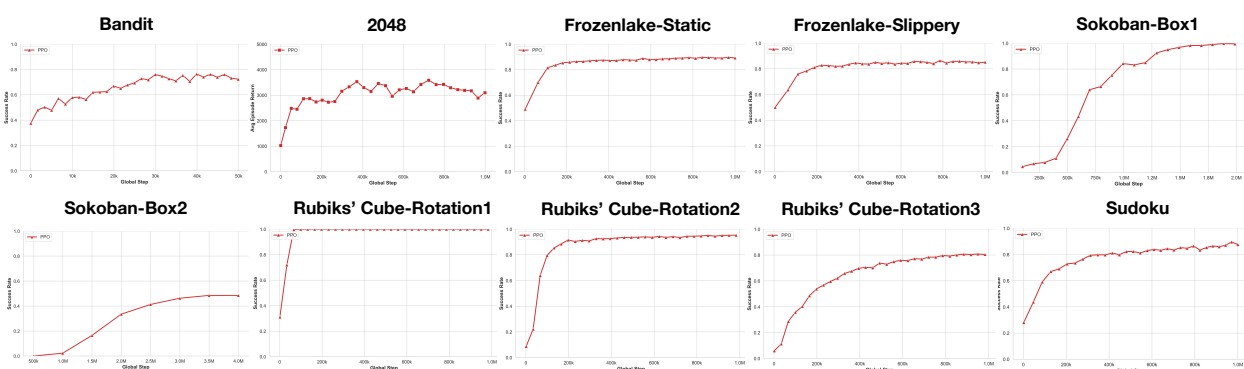

*Figure 5.* Scout-PPO detailed performance on 6 unseen tasks.

*Table 9.* Multi-task Agent via Sequential RL

| Model/Method | Bandit | FrozenLake | | Sokoban | | Rubiks' Cube | | | Sudoku | *Average* |
|---|---|---|---|---|---|---|---|---|---|---|
| | | *Static* | *Slippery* | *Box1* | *Box2* | *Rotation1* | *Rotation2* | *Rotation3* | | |
| Sequential RL with SCOUT | | | | | | | | | | |
| Qwen2.5-3B-It | 0.77 | 0.24 | 0.33 | 0.13 | 0.02 | 0.14 | 0.04 | 0.04 | 0.00 | 0.19 |
| +Exploration & Distillation Stage | 1.0 | 0.91 | 0.87 | 0.46 | 0.15 | 1.0 | 1.0 | 0.88 | 0.38 | 0.74 ↑+0.55 |
| +PPO on Bandit | 1.0 | 0.91 | 0.86 | 0.46 | 0.15 | 1.0 | 1.0 | 0.88 | 0.40 | 0.74 ↑+0.00 |
| +PPO on FrozenLake | 1.0 | 0.93 | 0.90 | 0.50 | 0.15 | 1.0 | 1.0 | 0.88 | 0.43 | 0.75 ↑+0.01 |
| +PPO on Sokoban | 1.0 | 0.89 | 0.88 | 0.93 | 0.59 | 1.0 | 1.0 | 0.86 | 0.48 | 0.85 ↑+0.10 |
| +PPO on Rubiks' Cube | 1.0 | 0.89 | 0.88 | 0.95 | 0.59 | 1.0 | 1.0 | 0.88 | 0.52 | 0.86 ↑+0.01 |
| +PPO on Sudoku | 1.0 | 0.89 | 0.88 | 0.95 | 0.59 | 1.0 | 1.0 | 0.89 | 0.98 | 0.91 ↑+0.05 |
| Sequential RL | | | | | | | | | | |
| Qwen2.5-3B-It | 0.77 | 0.24 | 0.33 | 0.13 | 0.02 | 0.14 | 0.04 | 0.04 | 0.00 | 0.19 |
| +PPO on Bandit | 0.86 | 0.26 | 0.25 | 0.14 | 0.02 | 0.11 | 0.04 | 0.04 | 0.00 | 0.19 ↑+0.00 |
| +PPO on FrozenLake | 0.84 | 0.22 | 0.30 | 0.17 | 0.06 | 0.23 | 0.05 | 0.08 | 0.00 | 0.22 ↑+0.03 |
| +PPO on Sokoban | 0.82 | 0.51 | 0.39 | 0.40 | 0.10 | 0.17 | 0.10 | 0.07 | 0.00 | 0.28 ↑+0.06 |
| +PPO on Rubiks' Cube | 0.70 | 0.52 | 0.50 | 0.37 | 0.09 | 0.33 | 0.18 | 0.11 | 0.02 | 0.31 ↑+0.03 |
| +PPO on Sudoku | 0.80 | 0.59 | 0.48 | 0.34 | 0.10 | 0.33 | 0.22 | 0.11 | 0.34 | 0.37 ↑+0.06 |

*Table 10.* SCOUT General Capability Evaluation on NLP Benchmarks

| Model/Task | MMLU | BoolQ | GSM8K | ARC-c | ARC-e | NQ-open | PIQA | TriviaQA | WinoG | *Average* |
|---|---|---|---|---|---|---|---|---|---|---|
| Qwen2.5-3B-it | 0.65 | 0.79 | 0.64 | 0.45 | 0.76 | 0.01 | 0.77 | 0.27 | 0.69 | 0.56 |
| + SCOUT | 0.64 | 0.83 | 0.65 | 0.51 | 0.81 | 0.10 | 0.78 | 0.39 | 0.69 | 0.60 ↑+0.04 |

## F. Used Prompts

In this section, we introduce the detailed System Prompts, State Estimation Prompts that we used in this paper. We follow SPA (Chen et al., 2025) on their state estimation prompts, and introduce new ones for Bandit, Rubiks' Cube and 2048.

### F.1. System Prompts

---
**System Prompt: Bandit**

- You are playing a bandit game. Goal: Maximize your total reward by choosing which arm to pull.
  Game Rules:
  1. There are 2 arms, named $\{name_a\}$ and $\{name_b\}$
  2. Each arm has its own reward distribution, related to their names.
  3. Analyze the symbolic meaning of each arm's name to guess how their reward distribution might behave.
  4. Based on the symbolic meaning of their names, which arm do you think is more likely to give higher rewards on average? Choose between $\{name_a\}$ and $\{name_b\}$, and output like <answer> $\{name_a\}$ </answer> or <answer> $\{name_b\}$ </answer>.
  Always output: <think> [Your thoughts] </think> <answer> [your answer] </answer>with no extra text. Strictly follow this format.

---
**System Prompt: 2048**

- You are playing the 2048 game on a 4x4 grid. Merge equal tiles by sliding Up, Right, Down, or Left.
  If a move is invalid (no tiles move), a small penalty is applied. Respond with a single action.
  Example: <answer>Up</answer>
  Always output: <think> [Your thoughts] </think> <answer> [your answer] </answer>with no extra text. Strictly follow this format.

---
**System Prompt: FrozenLake-Static**

- You are solving the FrozenLake puzzle. The observation includes both a symbol grid and zero-indexed coordinates for the start, goal, player, and any holes.
  Coordinates range from the top-left corner (0, 0) to the bottom-right corner (5, 5).
  Respond with a sequence of actions such as <answer>Left || Up || Up</answer>.
  Always output: <think> [Your thoughts] </think> <answer> [your answer] </answer>with no extra text. Strictly follow this format.

---
**System Prompt: FrozenLake-Slippery**

- You are solving the FrozenLake puzzle. The observation includes both a symbol grid and zero-indexed coordinates for the start, goal, player, and any holes.
  Coordinates range from the top-left corner (0, 0) to the bottom-right corner (5, 5).
  Beware that the ice is slippery, so the agent might slide and end up in an unintended tile.
  Respond with a sequence of actions such as <answer>Left || Up || Up</answer>.
  Always output: <think> [Your thoughts] </think> <answer> [your answer] </answer>with no extra text. Strictly follow this format.

---

**System Prompt: Sokoban**

- You are solving the Sokoban puzzle. You are the player and you need to push all boxes to targets.
  You are provided with a symbol grid and the zero-indexed coordinates of the player, each box, and each target.
  Coordinates range from the top-left corner (0, 0) to the bottom-right corner (5, 5).
  When you are exactly next to a box, you can push it by moving in the same direction.
  You cannot push a box through a wall, and you cannot pull a box.
  The answer should be a sequence of actions, like <answer>Right || Right || Up</answer>.
  Always output: <think> [Your thoughts] </think> <answer> [your answer] </answer>with no extra text. Strictly follow this format.

---

**System Prompt: Rubiks' Cube**

- You are solving a 2x2 Rubik's Cube (Pocket Cube). The goal is to restore the cube so that each of the faces consists of a single, unique color.
  Available actions use standard Singmaster notation for face rotations: U, U', D, D', L, L', R, R', F, F', B, B'.
  - Faces: U (Up), D (Down), L (Left), R (Right), F (Front), B (Back).
  - Modifiers: A letter alone means 90° clockwise (e.g., 'R'). A letter with prime (') means 90° counter-clockwise (e.g., "R'").
  Respond with a sequence of actions separated by "||".
  Example: <answer>U</answer>
  Always output: <think> [Your thoughts] </think> <answer> [your answer] </answer>with no extra text. Strictly follow this format.

---

**System Prompt: Sudoku**

- You are solving a Sudoku puzzle. Fill in the grid so that every row, column, and 2x2 box contains the numbers 1-4 without repetition.
  Initial cells are shown in [brackets] and cannot be modified. Empty cells are shown as dots (.).
  Place numbers one at a time using the format, for example: <answer>place 1 at row 2 col 3</answer> or <answer>1,2,3</answer>
  The environment will provide feedback on valid/invalid moves and show conflicts if any occur.
  Always output: <think> [Your thoughts] </think> <answer> [your answer] </answer>with no extra text. Strictly follow this format.

## F.2. State Estimation Prompts

**State Estimation Prompt: Bandit**

```
ou are playing a bandit game. Goal: Maximize your total reward by choosing which arm
 to pull.
Game Rules:
1. There are 2 arms, named $\{name_a\}$ and $\{name_b\}$.
2. Each arm has its own reward distribution, related to their names.
3. You need to analyze the symbolic meaning of each arm's name to guess their reward
 potential.

Example answer format:
<think>
<observation>
Arm B is named $\{name_b\}$. Symbolically, this implies analysis of name B.
</observation>
Based on the analysis, $\{name_a\}$ seems to represent risk/low value, while $\{
name_b\}$ implies wealth/stability.
<prediction>
If I pull $\{name_b\}$, I expect a higher average reward because [reasoning].
</prediction>
</think>
<answer> $\{name_b\}$ </answer>
```

```
A sample full output is as follows:
<think>
<observation>
Arm A is named "Rotten Apple". Symbolically, this implies decay and zero value.
Arm B is named "Golden Chalice". Symbolically, this implies treasure and high value.
</observation>
Comparing the two, the Golden Chalice is clearly superior in potential value.
<prediction>
Pulling "Golden Chalice" will likely yield a high positive reward, whereas "Rotten
Apple" might give zero or negative reward.
</prediction>
</think>
<answer> Golden Chalice </answer>
```

---

**State Estimation Prompt: 2048**

```
ou are playing the 2048 game on a 4x4 grid.
Merge equal tiles by sliding Up, Right, Down, or Left.
Use a zero-indexed grid where (0,0) is top-left and (3,3) is bottom-right.
If a move is invalid (no tiles move), a small penalty is applied.

Example answer format:
<think>
<observation>
[Current Grid Symbol Representation]
Tile values and positions: 2 at (0,0), 2 at (0,1), 4 at (3,3).
</observation>
I want to merge the two 2s in the top row. Sliding Right will merge them at (0,3).
<prediction>
[Predicted Grid Symbol Representation after move]
Predicted changes: The 2 at (0,0) and 2 at (0,1) merge into a 4 at (0,3). (3,3)
remains 4.
</prediction>
</think>
<answer>Right</answer>

A sample full output is as follows:
<think>
<observation>
2 2 . .
. . . .
. . . .
. . . 4
Non-empty tiles: 2 at (0,0), 2 at (0,1), 4 at (3,3).
</observation>
Checking moves:
Up: No merge.
Right: (0,0) and (0,1) are equal (2), they will merge to the right.
<prediction>
. . . 4
. . . .
. . . .
. . . 4
Merged tile 4 created at (0,3). Tile at (3,3) stays.
</prediction>
</think>
<answer>Right</answer>
```

State Estimation Prompt: Rubiks' Cube

```
ou are solving a 2x2 Rubik's Cube (Pocket Cube).
The goal is to restore the cube so that each of the faces consists of a single,
unique color.
Available actions: U, U', D, D', L, L', R, R', F, F', B, B'.
State representation format: A string of colors representing faces U, D, L, R, F, B.

Example answer format:
<think>
<observation>
Current State: [Color String or Grid]
Face U (Up): W W W W
Face F (Front): R R G G
Face R (Right): B B O O
...
</observation>
I need to fix the Front face top row. Rotating U (Up) clockwise will shift the top
row of F, R, B, L to the left.
<prediction>
Predicted State: [Predicted Color String or Grid]
Face F top row becomes the colors from Face R.
Face R top row becomes the colors from Face B.
...
</prediction>
</think>
<answer>U</answer>

A sample full output is as follows:
<think>
<observation>
      W W
      W W
O O   G G   R R   B B
O O   G G   R R   B B
      Y Y
      Y Y
Face U: All White. Face F: Green. Face R: Red. Face B: Blue. Face L: Orange. Face D:
 Yellow.
Cube is currently solved.
</observation>
To disrupt the state for demonstration, I will apply R (Right face clockwise).
This will move columns: F right col -> U right col -> B left col (inverted) -> D
right col -> F right col.
<prediction>
      W Y
      W Y
O O   G G   W W   B B
O O   G G   W W   B B
      Y R
      Y R
Face R rotates clockwise (colors cycle). Right column of F becomes Yellow (from D).
</prediction>
</think>
<answer>R</answer>
```

# G. Other Details

In this section, we show the the Scout Architecture and the Textualizer ($\Phi$) we use in this paper. We give the Textualizer of Sudoku as an example.

## G.1. Scout Architecture

*Listing 1.* Scout-DQN Architecture: MLP

```
1  class DQN_QNetwork(nn.Module):
2      def __init__(self, obs_dim: int, act_dim: int, hidden: int):
3          super().__init__()
4          self.net = nn.Sequential(
5              layer_init(nn.Linear(obs_dim, hidden)),
6              nn.ReLU(),
7              layer_init(nn.Linear(hidden, hidden)),
8              nn.ReLU(),
9              layer_init(nn.Linear(hidden, act_dim), std=0.01),
10         )
```

*Listing 2.* Scout-DQN Architecture: CNN

```
1  class DQN_QConv(nn.Module):
2      def __init__(self, obs_shape: Tuple[int, int, int], act_dim: int, dueling: bool =
   True):
3          super().__init__()
4          c, h, w = obs_shape
5          self._act_dim = act_dim
6          self.features = nn.Sequential(
7              layer_init(nn.Conv2d(c, 32, 3, 1, 1)),
8              nn.ReLU(),
9              layer_init(nn.Conv2d(32, 64, 3, 1, 1)),
10             nn.ReLU(),
11             layer_init(nn.Conv2d(64, 64, 3, 1, 1)),
12             nn.ReLU(),
13             nn.Flatten(),
14         )
15         fc_in = 64 * h * w
16         self.head = nn.Sequential(
17             layer_init(nn.Linear(fc_in, 512)),
18             nn.ReLU(),
19             layer_init(nn.Linear(512, act_dim), std=0.01),
20         )
```

*Listing 3.* Scout-PPO Architecture: MLP

```
1  class PPOAgent(nn.Module):
2      def __init__(self, envs):
3          super().__init__()
4          obs_shape = int(np.array(envs.single_observation_space.shape).prod())
5          hidden = 64
6          self.critic = nn.Sequential(
7              layer_init(nn.Linear(obs_shape, hidden)),
8              nn.Tanh(),
9              layer_init(nn.Linear(hidden, hidden)),
10             nn.Tanh(),
11             layer_init(nn.Linear(hidden, 1), std=1.0),
12         )
13         self.actor = nn.Sequential(
14             layer_init(nn.Linear(obs_shape, hidden)),
15             nn.Tanh(),
16             layer_init(nn.Linear(hidden, hidden)),
17             nn.Tanh(),
18             layer_init(nn.Linear(hidden, envs.single_action_space.n), std=0.01),
19         )
```

*Listing 4.* Scout-PPO Architecture: CNN

```
1
2  class CNN_Agent(nn.Module):
3      def __init__(self, envs):
4          super().__init__()
5          c, h, w = envs.single_observation_space.shape
6          hidden = 1024
7          self.net = nn.Sequential(
8              layer_init(nn.Conv2d(c, 64, kernel_size=3, stride=1, padding=1)),
9              nn.ReLU(),
10             layer_init(nn.Conv2d(64, 128, kernel_size=3, stride=1, padding=1)),
11             nn.ReLU(),
12             layer_init(nn.Conv2d(128, 128, kernel_size=3, stride=1, padding=1)),
13             nn.ReLU(),
14             nn.Flatten(),
15             layer_init(nn.Linear(128 * h * w, hidden)),
16             nn.ReLU(),
17         )
18         self.critic = layer_init(nn.Linear(hidden, 1), std=1.0)
19         self.actor = layer_init(nn.Linear(hidden, envs.single_action_space.n), std
    =0.01)
```

### G.2. Textualizer

In this section, we provide a concrete example of the Textualizer that transforms scout trajectories $D_{scout}$ into language-based trajectories $D_{LLM}$. Since the original task environments are standard Gym-style environments, the corresponding language-based environments are constructed by expressing environment states, feedback, and actions as natural language descriptions. To ensure that the scouts interact with exactly the same underlying environments, we directly train the scouts in the original Gym-style tasks, without any language augmentation.

Moreover, as the RAGEN codebase (Wang et al., 2025d) already provides canonical language descriptions for all tasks, the transformation from $D_{scout}$ to $D_{LLM}$ is implemented by deterministically substituting the symbolic states and actions in these predefined templates with those observed in $D_{scout}$. This process performs a direct serialization of existing information and does not introduce additional task structure, transition rules, or planning heuristics.

*Table 11.* **Mapping from Scout Trajectories to Language Trajectories via Textualizer (Φ).** This table demonstrates the full mapping process. We take Sokoban as an example. The **left column** represents the trajectories the collected $D_{scout}$. The **right column** represents the structured language trajectories $D_{LLM}$ transferred from the $D_{scout}$ by the Textualizer.

---

**Example: Sokoban**

| Scout Trajectories | Language Trajectories |
|---|---|
| *State* 
 ```###### ``` 
 ```###__# ``` 
 ```###X_# ``` 
 ```#_#OP# ``` 
 ```#____# ``` 
 ```###### ``` | *State* 
 **SYSTEM INSTRUCTION:** 
 You are solving the Sokoban puzzle. You are the player and you need to push all boxes to targets. You are provided with a symbol grid and the zero-indexed coordinates of the player, each box, and each target. Coordinates range from the top-left corner (0, 0) to the bottom-right corner (5, 5). When you are exactly next to a box, you can push it by moving in the same direction. You cannot push a box through a wall, and you cannot pull a box. The answer should be a sequence of actions, like `<answer>Right || Right || Up</answer>`. 

 **Current Turn:** 
 Turn 1: 
 State: 
 Grid Map: 
 ```###### ``` 
 ```###__# ``` 
 ```###X_# ``` 
 ```#_#OP# ``` 
 ```#____# ``` 
 ```###### ``` |
| *Action* 
 ```1``` | *Action* 
 ```<think></think><answer>Down</answer>``` |
| *Reward* 
 ```0.0``` | *Reward* 
 ```Reward:``` 
 ```0.0``` |
| *Augmentations* 
 ```NaN``` | *Augmentations* 
 You have x actions left. Always output: `<think>[Your thoughts]</think><answer>[your answer]</answer>` with no extra text. Strictly follow this format. Max response length: n words (tokens). |

