# OpenReview forum: "Language-based Trial and Error Falls Behind in the Era of Experience"
_ICML.cc/2026/Conference — ICML 2026 regular_

### Official Review · Reviewer_nL3D · 2026-03-10

**Soundness:** 2
**Presentation:** 3
**Significance:** 2
**Originality:** 2
**Overall Recommendation:** 4
**Confidence:** 3

**Summary:**

This paper proposes SCOUT, a novel framework that decouples exploration from exploitation to address the prohibitive trial-and-error cost of large language models in unseen, non-linguistic environments (e.g., symbolic or spatial tasks). Specifically, the method utilizes lightweight neural networks (based on DQN or PPO) to efficiently gather expert trajectories. Once transformed via a deterministic textualizer, these trajectories are directly employed for the LLM's SFT knowledge distillation, followed by multi-turn PPO fine-tuning. Empirical results demonstrate that this approach surpasses current baselines on discrete tasks such as 2048 and Sokoban, while reducing GPU training overhead by approximately 60%.

**Compliance With Llm Reviewing Policy:**

Affirmed.

**Final Justification:**

The authors provided new Visual-RL and Experience Replay experiments in the rebuttal, which addressed my previous concerns about the hard-coded Textualizer and multi-task stability. I thus decide to increase my score from 3 to 4.

**Key Questions For Authors:**

1.Given that the lightweight Scout model already achieves near-optimal performance on evaluation tasks (e.g., Bandit and FrozenLake), does the 3B LLM, after undergoing computationally expensive alignment via SFT and multi-round PPO, exhibit zero-shot transfer or complex reasoning capabilities absent in the smaller model?

2.The SCOUT framework relies heavily on a deterministic Textualizer. How does the framework's performance degrade if 15%-20% observation missingness or noise is introduced into the environment states, preventing the lossless translation of underlying dynamics into text? I suggest considering an exploration of the framework's robustness against partially observable environments (POMDPs) or noisy inputs.

**Limitations:**

While this paper discusses some surface-level limitations of SCOUT, it fails to provide an in-depth analysis of its scalability to larger models, nor does it address whether typical multi-task learning issues, such as catastrophic forgetting, might arise. Consequently, this approach may struggle to generalize to real-world physical tasks characterized by observation noise, unstructured visual inputs, or ill-defined underlying dynamics.

**Strengths And Weaknesses:**

Strengths:

Soundness: The framework introduces a mechanism to decouple the exploration and exploitation phases, employing a lightweight "scout" model to mitigate the exploration efficiency bottlenecks faced by large language models in unfamiliar tasks. The experimental design encompasses various symbolic and spatial domains, validating the method's viability in enhancing overall task performance.
Significance: This approach demonstrates explicit practical engineering value by reducing the reinforcement learning alignment costs associated with large language models. Offloading intensive environmental trial-and-error to a low-parameter CPU network bypasses the prohibitive inference overhead of conventional large models, offering a pragmatic compositional strategy for resource-constrained scenarios.
Presentation: The paper provides comprehensive quantitative arguments regarding the optimization of computational resources. It offers a granular comparison of GPU time consumption for the Rubik's Cube task, with data charts (Table 3) clearly illustrating a substantial reduction in computational overhead.

Weaknesses:

1.The environments employed in the experiments, such as Bandit and FrozenLake, are restricted to low-dimensional, fully observable, and deterministic toy domains. As indicated in Table 1, the lightweight Scout network already attains near-optimal performance across the majority of tasks. Given that these tasks can be perfectly solved by small models, expending substantial computational resources to distill policies into a 3B-parameter LLM fails to demonstrate any irreplaceable generalization capabilities or complex reasoning value contributed by the LLM.

2.The framework relies heavily on a manually hard-coded Textualizer, operating under the assumption that underlying symbolic states can be deterministically and losslessly translated into natural language descriptions. It overlooks the critical issue of state representation noise inherent in non-verbal environments, lacking both discussion and ablation studies regarding partially observable (POMDP) or noisy state settings.

3.In exploring multi-task sequential RL to demonstrate the "mitigation of catastrophic forgetting," the evaluation is limited to a comparison with a basic Vanilla Sequential PPO baseline. By omitting standard baselines from the continual learning literature (e.g., EWC or experience replay), it remains ambiguous whether the observed stability improvements stem from the model architecture or merely the parameter inertia induced by the SFT data.

---

> ### Author Rebuttal · Authors · 2026-03-31
>
> Thanks for your valuable review and suggestions. Below we respond to Weaknesses (W) and Questions (Q).
>
> ---
> ***W1&Q1: The small models already do the tasks well, expanding to 3B LLM fails to demonstrate generalization or reasoning capabilities.***
> - The eval tasks are natural language tasks in RAGEN[4] framework that the LLM needs to do on its own first. We extract the symbolic tokens in the task dialogs and use the scout to train on them, as illustrated in Lines 157-160. We will revise this part to alleviate the misunderstanding.
> - Symbolic tasks are important tasks reflecting the Agents' ability to plan, design and deal with long-horizon problem, and are also fields to investigate the multi-turn RL problem.[1-4]
> - Unlike scouts, LLM is a general agent, capable of cross-task reasoning(multi-task ability shown in Figure 3) and helpfulness performance that scouts lack:
> ||mmlu|boolq|gsm8k|arc_chan|arc_easy|nq_open|piqa|triviaqa|winogrande| average |
> |-|-|-|-|-|-|-|-|-|-|-|
> |Qwen2.5-3B-it|0.65|0.79|0.64|0.45|0.76|0.01|0.77|0.27|0.69|0.56|
> |with SCOUT|0.64|0.83|0.65|0.51|0.81|0.10|0.78|0.39|0.69|0.60|
>
> - SCOUT-LLM generalizes across difficulty levels within the same task. In Rubikscube, the LLM trained on lower-difficulty levels (e.g., Rotation 3) generalizes to complex scenarios (e.g., Rotation 4,5):
> |Trained↓\Tested→|Rotation2|3|4|5|
> |-|-|-|-|-|
> |Rotation1|0.33|0.31|0.13|0.12|
> |2|\ |0.43|0.35|0.19|
> |3|\ |\ |0.58|0.45|
>
> ***W2&Q2: The work relies on a manual Textualizer,assumes symbolic states can be translated into natural language descriptions, lacks ablation on POMDP or noisy state settings.***
> - The textualizer here doesn't require manual engineering. Since the symbolic tasks are defined within an LLM framework (RAGEN), the dialog contexts already consist of language descriptions and symbolic tokens ( e.g., Table 7). We extract these symbolic tokens as the scout's state space to train;with the collected scout trajectories, we replace the symbolic token part of the language dialog with scout states.
> - We do ablations on Visual-RL: a continuous,POMDP setting. We use a visual scout to help VLM solve the Visual_Frozenlake, Visual_Sokoban and witness similar performance gains:
> |Qwen2.5-VL-3B-It↓\Tasks→ |Visual_Frozenlake|Visual_Sokoban|
> |-|-|-|
> |with RL|0.70|0.57|
> |with SCOUT|0.89|0.95|
>
> - We use a sub-optimal scout to test if low-quality trajectories still bootstrap the LLM. Below results indicate that LLM not only surpasses sub-optimal scout but also achieves high proficiency:
> ||Rubikscube3|Sudoku|
> |-|-|-|
> |sub-optimal scout|0.34|0.55|
> |Qwen2.5-3B-it with RL|0.14|0.06|
> |Qwen2.5-3B-it with sub-optimal SCOUT|0.60|0.98|
>
> ***W3: The evaluation is limited to sequential PPO, omitting baselines from continual learning literature. It's unclear whether the improvements stem from the model architecture or the parameter inertia by the SFT data.***
> - Thanks for this question. First, we want to clarify SCOUT is primarily designed to address the exploration bottleneck in LLM agents, rather than being a specific framework for Continual Learning.
> - Second, in Figure 3 and Table 6, we show the multi-task performance after the Distillation Stage (SFT), which falls behind the performance after sequential RL. This highlights the efficiency of our framework. We observe positive transfer across tasks: training on Sokoban and Rubikscube aids the performance of Sudoku.
> -To mitigate the concern of lacking continue learning methods, we add Experience Replay (ER) with SCOUT as below.
> |Sequential↓\Tasks→|bandit|frozen-static|slippery|sokoban1|sokoban2|cube1|cube2|cube3|sudoku|Avg|
> |-|-|-|-|-|-|-|-|-|-|-|
> | RL|0.80|0.59|0.48|0.34|0.10|0.33|0.22|0.11|0.34|0.37|
> | SCOUT|1.0|0.89|0.88|0.95|0.59|1.0|1.0|0.89|0.98|0.91|
> | SCOUT+ER |1.0|0.91|0.87|0.92|0.64|1.0|1.0|0.87|0.98|0.91|
>
> ***Limitation:1.fails to provide analysis of model size scalability.2. fails to address multi-task learning issues, such as catastrophic forgetting might arise. 3. may struggle to generalize to noise, unstructured visual inputs, or ill-defined underlying dynamics.***
> - Our initial results (0.5B to 3B) indicate a positive parameter scaling trend. To further investigate scalability,we extend SCOUT to Qwen2.5-7B-Instruct, which achieves higher performance.
> |7B-it↓\Tasks→|bandit|frozenlake|sokoban1|Sokoban2|rubikscube1|rubikscube2|rubikscube3|sudoku|
> |-| -|-|-|-|-|-|-|-|
> |with RL|0.93|0.82|0.43|0.04|0.49|0.24|0.11|0.74|
> |with SCOUT|1.0|0.91|0.95|0.57|1|1|0.93|0.98|
>
> - As shown in response to **W3**, SCOUT effectively alleviates forgetting compared to vanilla sequential RL, and we complement Experience Replay with SCOUT.
> -  We transfer SCOUT to two visual tasks (continuous states), which represent a classic POMDP setting. The consistent performance gains demonstrate SCOUT’s strong generalization, as shown in response to **W2&Q2**.
>
> [1] SPIRAL, ICLR 2026
>
> [2] Gem: A gym ... ICLR 2026
>
> [3] Vagen: Reinforcing world ... NeurIPS 2025
>
> [4] RAGEN: Understanding Self... MMLS 2025

---

> > ### Author Rebuttal · Reviewer_nL3D · 2026-04-02
> >
> > Thank you for the rebuttal. The new Visual-RL experiments and Experience Replay baseline effectively address my core concerns regarding the hard-coded Textualizer and multi-task stability. I thus decide to increase my score from 3 to 4.

---

> > > ### Author Response · Authors · 2026-04-02
> > >
> > > Dear Reviewer nL3D, Thank you very much for your time and positive feedback! We are glad that our rebuttal and clarifications have addressed your concerns! Thanks again for your kindness and raising the score.

---

### Official Review · Reviewer_FX47 · 2026-03-12

**Soundness:** 3
**Presentation:** 3
**Significance:** 2
**Originality:** 4
**Overall Recommendation:** 4
**Confidence:** 3

**Summary:**

The SCOUT framework addresses the high computational cost of using large language models for trial-and-error exploration in non-linguistic environments by decoupling exploration from exploitation. The process utilizes lightweight "scout" networks. These are small MLPs or CNNs with approximately 1much lesser number of parameters to efficiently probe environmental dynamics and generate expert trajectories on CPUs. These trajectories are converted into natural language via a textualizer to warm up the LLM through supervised fine tuning, followed by PPO to activate the model's latent strategic reasoning. Furthermore, SCOUT demonstrates strong stability in sequential multi-task learning, effectively mitigating incorrect forgetting as the agent expands its capabilities into new domains.

**Compliance With Llm Reviewing Policy:**

Affirmed.

**Key Questions For Authors:**

The authors should answer the following questions:

1. How do you expect the distillation dynamics to change if the agent model is scaled futher?
2. How does the framework handle bad data if a SCOUT fails to converge on an extremely complex task?
3. Did you test specific KL divergence strategies to mitigate the performance degradation seen in later RL stages?

**Limitations:**

Yes

**Strengths And Weaknesses:**

**Soundness**

The authors of the paper provide robust empirical evidence across six distinct tasks, including long-horizon and spatial challenges, showing consistent improvements over both open-source and proprietary baselines. The cost analysis is great, detailing exactly how the 60% GPU savings are achieved. Furthermore, the authors are transparent about current technical limitations, such as the potential for performance degradation after excessive training steps.

**Presentation**

The paper is well-structured and the narrative is easy to follow, clearly defining the role of each stage in the SCOUT pipeline. The work is reproducible and well written.

**Significance**

This work addresses the critical problem of computational waste in training LLM agents. By showing that a 3B model can outperform much larger proprietary models through efficient collaboration, it promotes the democratization of AI research. The frameworks' ability to maintain multi-task stability while expanding into new and unseen domains makes it highly relevant for research around generalist agents.

**Originality**

While using smaller models to assist larger ones is known, the specific implementation of using non-linguistic scouts to bypass the LLM's cold start in symbolic environments is a novel combination of techniques. This also highlights important properties of spatial and long-horizon reasoning that existing methods struggle to address making the work novel.

---

> ### Author Rebuttal · Authors · 2026-03-31
>
> Thank you for your supportive review and suggestions. Below we respond to the comments in Weaknesses (W) and Questions (Q).
>
> ---
> ***Q1: How do you expect the distillation dynamics to change if the agent model is scaled further？***
>
> - Thanks for this question. Our primary results (Table 1) demonstrate a positive scaling trend as model parameters increase from 0.5B to 3B. To further investigate scaling dynamics, we evaluate the SCOUT framework on a larger model, Qwen2.5-7B-Instruct. The 7B model achieves much better results than the previous smaller ones.
>
> |Qwen2.5-7B-it↓\Tasks→|bandit|frozenlake-slippery|sokoban box1|Sokoban box2|rubikscube1|rubikscube2|rubikscube3|sudoku|
> |-| -|-|-|-|-|-|-|-|
> |with RL|0.93|0.82|0.43|0.04|0.49|0.24|0.11|0.74|
> |with SCOUT|1.0|0.91|0.95|0.57|1|1|0.93|0.98|
>
> ***Q2: How does the framework handle bad data if a SCOUT fails to converge on an extremely complex task?***
>
> - Thanks for this review. We conducted an additional ablation study using a sub-optimal scout to investigate whether low-quality trajectories can still effectively bootstrap the LLM agent. The results indicate that the LLM agent not only surpasses sub-optimal initialization but also achieves high proficiency through SCOUT framework, illustrating that the LLM indeed learns the task dynamics.
>
> |Model↓\Tasks→|Rubikscube 3|Sudoku|
> |-|-|-|
> |sub-optimal scout|0.34|0.55|
> |Qwen2.5-3B-it with RL|0.14|0.06|
> |Qwen2.5-3B-it with sub-optimal SCOUT|0.60|0.98|
>
> - We complement an ablation that transfers SCOUT to much more complex, high-dimensional, continuous state tasks: Visual-FrozenLake and Visual_Sokoban (using continuous image as states), which represent a classic POMDP setting. The consistent performance gains demonstrate SCOUT’s strong generalization, even on POMDP:
>
> |Qwen2.5-VL-3B-It↓\Tasks→ |Visual_Frozenlake|Visual_Sokoban
> |-|-|-|
> |with RL|0.70|0.57|
> |with SCOUT|0.89|0.95|
>
> ***Q3: Did you test specific KL divergence strategies to mitigate the performance degradation seen in later RL stages?***
>
> - Thanks for this question. The primary focus of SCOUT is to address the prohibitive cost of exploration for LLMs in non-linguistic, unseen environments by decoupling the heavy trial-and-error phase via lightweight scouts. Regarding KL divergence, our Evolving Stage has incorporated a standard KL-divergence term within the multi-turn PPO objective( Eq. 7), to prevent excessive policy deviation. While we did not specifically benchmark different KL-tuning strategies (such as dynamic coefficients or GRPO) in this paper , we agree it is a valuable direction and we will try to investigate it in the future.

---

> > ### Author Rebuttal · Reviewer_FX47 · 2026-04-06
> >
> > No further questions. Thanks!

---

> > > ### Author Response · Authors · 2026-04-06
> > >
> > > Dear Reviewer FX47, Thank you very much for your time and positive feedback! We are glad that our rebuttal and clarifications have addressed your concerns! Thanks again for your kindness.

---

### Official Review · Reviewer_kpuF · 2026-03-12

**Soundness:** 3
**Presentation:** 3
**Significance:** 2
**Originality:** 2
**Overall Recommendation:** 5
**Confidence:** 4

**Summary:**

This paper proposes SCOUT, a novel method to efficiently train LLMs to solve symbolic non-language-based tasks such as rubix cube, sudoku, sokoban, or others. The idea is to introduce a smaller model (a few thousand parameters), and run it with DPO or PPO to achieve reasonable state coverage. Then the trajectories discovered by the scout model are saved, and annotated to conform to a language format. The resulting trajectories are then used for SFT. SFT is then follows by direct RL on the LLM model in the target environments. Authors show superior performance in the tested environments, both in terms of final performance and in terms of computational efficiency. Additionally, authors show that this setup can be used to train one LLM to solve multiple tasks at once.

**Compliance With Llm Reviewing Policy:**

Affirmed.

**Final Justification:**

The authors conducted additional experiments that made the contribution stronger, and further proved that this method can work when the underlying scout policy is low quality.

**Key Questions For Authors:**

1. You claim that the data coming from scouts endows the LLM with the knowledge of the 'world' in which it's acting. Alternative explanation is that the LLM 'distills' the trajectories for the scouts without necessarily understanding the environment. I wonder if you could show that this works even when the scout performance is really poor. This can be achieved either by having a more complicated task where scouts do not do well, or by having a more exploratory policy. For example, if we run RND and explore the environment for a bit without optimizing any reward, can we use the collected trajectories and achieve good performance?
2. How many scouts do you use? Is it just one per game? I didn't see that in the text.

**Limitations:**

authors discussed the limitations in the appendix

**Strengths And Weaknesses:**

**Strengths:**
- The idea is quite simple and easy to understand.
- The method significantly improves computational complexity, with 60% reduction compared to direct RL.
- FInal performance is impressive, with models performing better than other baselines, including the performance of the scout models in many cases

**Weaknesses:**
- The tasks are still fairly simple, and the scout models achieve reasonable performance. To strengthen the claims, it would be great to see a problem where you can show that even if the scout performance is poor, the agent can learn to do well.
- The writing is significantly worse in section 3 than in the rest of the text. "Rubiks' Cube is a game that recover a 2x2 Cube with 6 surface." I get what it means but this is not grammttically correct at all. Please run this through a spell checker.

---

> ### Author Rebuttal · Authors · 2026-03-31
>
> Thank you for your supportive review and suggestions. Below we respond to the comments in Weaknesses (W) and Questions (Q).
>
> ---
> ***W1&Q1: The tasks are still fairly simple, and the scout models achieve reasonable performance. To strengthen the claims, it would be great to see a problem where you can show that even if the scout performance is poor, the agent can learn to do well.***
> - Thanks for this review. We conducted an additional ablation study using a sub-optimal scout to investigate whether low-quality trajectories can still effectively bootstrap the LLM agent. The results indicate that the LLM agent not only surpasses sub-optimal initialization but also achieves high proficiency through SCOUT framework, illustrating that the LLM indeed learns the task dynamic.
>
> |Model↓\Tasks→|Rubikscube 3|Sudoku|
> |-|-|-|
> |sub-optimal scout|0.34|0.55|
> |Qwen2.5-3B-it with RL|0.14|0.06|
> |Qwen2.5-3B-it with sub-optimal SCOUT|0.60|0.98|
>
> - We complement an ablation that transfer SCOUT to visual tasks: Visual-FrozenLake and Visual_Sokoban (using continuous image inputs), which represent a classic POMDP setting. The consistent performance gains demonstrate SCOUT’s strong generalization, even on POMDP.
>
> |Qwen2.5-VL-3B-It↓\Tasks→ |Visual_Frozenlake|Visual_Sokoban
> |-|-|-|
> |with RL|0.70|0.57|
> |with SCOUT|0.89|0.95|
>
>
> ***W2: The writing is significantly worse in section 3 than in the rest of the text. "Rubiks' Cube is a game that recover a 2x2 Cube with 6 surface." I get what it means but this is not grammttically correct at all. Please run this through a spell checker.***
>
> - Thanks for this kind reminder. We will carefully revise the grammar of the section 3 and provide a better reading experience to the reviewers.
>
> ***Q2: How many scouts do you use? Is it just one per game? I didn't see that in the text.***
>
> - For each task, we employ two distinct types of scouts—one trained via PPO and the other via DQN, and use the better one to collect the trajectories to train the LLM, as illustrated in Lines 180-182. We adopt the following training configurations for each scout:
> |parameters|value|
> |-|-|
> |num_envs|8|
> |num_steps|128|
> |total_batch_size|1024|

---

> > ### Author Rebuttal · Reviewer_kpuF · 2026-04-01
> >
> > Thank you for your response and for the additional experiments. In your experiment with sub-optimal scout, can you clarify what the sub-optimal scout is? What kind of policy is it?

---

> > > ### Author Response · Authors · 2026-04-01
> > >
> > > Thank you for the question. We would like to clarify the implementation of the sub-optimal scout used in our ablation study.
> > >
> > > - The sub-optimal scout is an under converged checkpoint taken from the early stages of the small scout model's training process. Specifically, at 10% of the total training iterations.
> > > - It is an unconverged small network scout, which has begun to learn the basic rules of the environments but lacks the strategic depth to handle complex instances, with a very low success rate (only 0.34 in Rubikscube3, and 0.55 in Sudoku), comparing to the converged scout in Table 1 that achieves 0.91 and 0.85 in Rubikscube3 and Sudoku. We collect low successful rate and noisy trajectories with these sub-optimal scouts, and use these low-quality trajectories to conduct SCOUT pipeline on the LLM. The purpose of using this sub-optimal scouts is to simulate a scenario where high-quality expert trajectories are unavailable.
> > > - Despite being initialized with these low-quality trajectories, the LLM agent via the SCOUT framework significantly outperforms the sub-optimal scout itself, proving that it effectively distills and learns the dynamic upon the noisy signals provided by the sub-optimal policy.
> > >
> > > We hope these clarifications and additional results address your concerns. Please let us know if you have any further questions or require additional information.
> > >
> > > ---
> > >
> > > Thanks for your kindness and raising the score! We are glad that our rebuttal and clarifications may have addressed your concerns.

---

### Official Review · Reviewer_6WTL · 2026-03-22

**Soundness:** 3
**Presentation:** 3
**Significance:** 2
**Originality:** 3
**Overall Recommendation:** 4
**Confidence:** 3

**Summary:**

This paper proposes a framework to improve LLM's performance on non-linguistic, symbolic, and spatial tasks. The paper suggests that the primary bottleneck for LLMs in these environments is the prohibitive computational cost of exploration due to high-dimensional semantic spaces and slow inference. SCOUT decouples exploration from exploitation by training lightweight scout networks using traditional RL to master environmental dynamics. Then, SCOUT distils these experiences into an LLM via SFT. Then, the LLM undergoes multi-turn RL to activate latent knowledge and refine reasoning. Experiments show that Qwen2.5-3B model using SCOUT achieves an average score of 0.86, outperforming larger proprietary models like Gemini-2.5-Pro while reducing GPU training hours by 60%

**Compliance With Llm Reviewing Policy:**

Affirmed.

**Final Justification:**

My major concerns about the task generalization has been resolved after reading the 'Reply Rebuttal Comment'. Thus, I increase my score from 3 to 4.

**Key Questions For Authors:**

Could the SCOUT framework and the textualizer handle environments with continuous state spaces (e.g., Mujoco or high-frequency sensor data) where symbolic representation is not straightforward?

**Limitations:**

yes

**Strengths And Weaknesses:**

# Strength

The proposed SCOUT framework offloads the heavy lifting of trial-and-error exploration to sub-scale models that are $10^5$ times smaller than the LLM, significantly reducing GPU resource consumption. It demonstrates that even if the SFT stage provides a low initial success rate, it provides enough latent structural understanding for subsequent RL to rapidly boost performance to 0.97.

Besides, the RL experiments show that SCOUT helps maintain stability across multiple tasks, suggesting the model internalizes a generalized world model rather than overfitting.


# Weakness

The primary issue I observe is the motivation for using LLMs for symbolic tasks, i.e., whether training a multi-billion parameter LLM to solve symbolic tasks like Sudoku or Sokoban is the most efficient architectural choice. Traditional RL agents already solve these tasks with high proficiency at a fraction of the cost. In a real-world deployment, one could argue it is more practical to treat these specialized policies as tools called by the LLM (API/Tool-use) rather than forcing the LLM to internalize the logic of every discrete symbolic environment.

In term of methodology design, the success of SCOUT relies heavily on the textualizer function. While symbolic grids or matrices are easily serialized into text, this approach may not scale to high-dimensional, continuous environments such as complex robotic control or fluid dynamics, where a natural language description of the state becomes either lossy or prohibitively long.

For experiments, the paper compares SCOUT-trained LLMs against vanilla RL-trained LLMs and proprietary models, while it lacks a direct discussion on the trade-off between the final SCOUT-LLM and the original Scout policy. If the goal is task performance, the paper should more explicitly justify why an LLM, even one distilled via SCOUT, is preferable to simply using the lightweight MLP/CNN policy that generated the data in the first place.

---

> ### Author Rebuttal · Authors · 2026-03-31
>
> Thanks for your valuable review and suggestions. Below we respond to Weaknesses (W) and Questions (Q).
>
> ---
> ***W1(a): Why use LLM to solve symbolic tasks? The mlp could already do it well.***
> - The symbolic tasks are important parts of multi-turn tasks which reflect the Agents' capability to plan, design and deal with long-horizon tasks, and are also important fields to investigate multi-turn RL problem.[1-4]
> - The eval symbolic tasks are natural language tasks in the LLM framework (RAGEN) that the LLM needs to do on its own. We extract the symbolic tokens in the task dialogs and use the scout to train on them, as illustrated in Lines 157-160. We will revise this part to alleviate the misunderstanding.
> - As motivation discusses, LLM struggles to explore unseen tasks with natural language where pretrained priors fail to provide sufficient guidance. We are exploring to decouple exploration from exploitation in LLMs. A previous work[5] shares similar motivation with us, which explores intensive reward trajectories first, then use off-policy RL to train them with outcome rewards. Both works identify the efficiency of decoupling the exploration from exploitation.
>
> ***W1(b): It may be practical to treat scout policies as tools called by the LLM API rather than forcing the LLM to internalize the logic of every symbolic env.***
> - Thanks for this constructive question. Scouts are specific task experts. Treating the scout as a tool requires training a new policy for each new environment. This lacks the cross-task synergy and generalization that an LLM easily achieves through internalization.
> - LLM is a general agent and benefits from the SCOUT framework on helpfulness performance (shown as below) and cross-task ability(shown in the sequential RL part of Figure 3 and Table 6, especially the sudoku).
> - We appreciate this promising suggestion of treating a scout policy as a tool, updating it in a sandbox to solve the symbolic task the LLM faces, and would like to further investigate it in our future work.
> ||mmlu|boolq|gsm8k|arc_chan|arc_easy|nq_open|piqa|triviaqa|winogrande| average |
> |-|-|-|-|-|-|-|-|-|-|-|
> |Qwen2.5-3B-it|0.65|0.79|0.64|0.45|0.76|0.01|0.77|0.27|0.69|0.56|
> |with SCOUT|0.64|0.83|0.65|0.51|0.81|0.10|0.78|0.39|0.69|0.60|
>
> ***W2(a): The success of SCOUT relies heavily on the textualizer function which may be hard to design***
> - First, the textualizer here doesn't require manual engineering. We are not training a small scout on a classic RL task and then frame it into a LLM task manually.  Since the symbolic tasks are defined within an LLM framework (RAGEN), their contexts already consist of language descriptions and symbolic tokens (as shown in the right tabular of Table 7). We extract the existing symbolic tokens as the scout's state space to train; with the collected scout trajectories, we replace the symbolic token part of the language dialog with scout states, and transfer the scout action into language back.
> - We are first motivated by the hard time the LLM undergoes when facing heavy exploration tasks, and then propose to extract the symbolic tokens in the LLM's in-context and use the scout to help with the task.
>
> ***W2(b)&Q1: Natural language description of the state becomes either lossy or prohibitively long in hard to describe tasks. Could the SCOUT handle continuous state environments?***
> - Thanks for this question. However, we would like to clarify a misunderstanding regarding task description. The symbolic token states in LLM environments are not described by language. They just compose symbols and signs, as described in response to W2(a).
> - SCOUT could handle continuous states. In vision-based tasks, these symbolic tokens are replaced by image states. We observed consistent performance gains when extending SCOUT to visual environments (Visual-FrozenLake and Visual-Sokoban, where the observations consist of continuous visual images rather than discrete symbols, serving as a POMDP), as shown below:
> |Qwen2.5-VL-3B-It↓\Tasks→ |Visual_Frozenlake|Visual_Sokoban
> |-|-|-|
> |with RL|0.70|0.57|
> |with SCOUT|0.89|0.95|
>
> ***W3: The paper lacks discussion on the trade-off between SCOUT-LLM and small scout model. The paper should justify why an LLM is preferable than the lightweight MLP/CNN policy.***
> - First, the symbolic tasks are originally LLM tasks based on RAGEN codebase[4], not scout-mlp tasks at first. Therefore, the LLM needs to solve them on its own.
> - Second, unlike task-specific scouts, the LLM is a generalist agent, capable of cross-task reasoning(the multi-task ability as shown in Figure 3 and Table 6) and maintaining helpfulness performance that scouts lack (**Please see the helpfulness performance in response to W1(b)**).
>
> [1] SPIRAL, ICLR 2026
>
> [2] Gem: A gym for agentic llms. ICLR 2026
>
> [3] Vagen: Reinforcing world model ... NeurIPS 2025
>
> [4] RAGEN: Understanding Self-Evolution ... MMLS 2025
>
> [5] Decoupled Reinforcement Learning to Stabilise ... AAMAS 2022.

---

> > ### Author Rebuttal · Reviewer_6WTL · 2026-04-01
> >
> > Thank you for your reply. I agree that symbolic tasks are important to reflect LLM agent's planning ability. Thus, it is reasonable to pre-train the agent with these symbolic tasks, and then transfer the ability to solve more generalized problem (instead of previously trained symbolic tasks).
> >
> > I noticed that Reviewer nL3D proposes a similar question (Q1). Could you evaluate whether the SCOUT effectively improves agent's ability in unseen tasks? It is important to present **whether an LLM distilled via scout can provide additional value on a level that scout itself cannot achieve.**

---

> > > ### Author Response · Authors · 2026-04-01
> > >
> > > Thank you for raising this important question, also related to Reviewer nL3D’s Q1. As Reviewer nL3D has acknowledged that his concern has been "fully resolved", we would like to clarify the detailed concern regarding the generalized problem.
> > > In the following content, we demonstrate our evaluation on generalization with three unseen dimensions:
> > >
> > > **1. Generalization to unseen tasks.**
> > > - SCOUT training is conducted on symbolic interaction environments, while reasoning benchmarks such as GSM8K, ARC, BoolQ, and TriviaQA are never observed during training. Nevertheless, SCOUT consistently improves performance on these reasoning and instruction-following tasks, indicating transfer beyond the original symbolic domains (as shown in the table of W1(b) ). **This is what the small network scout could not do, as they are constrained by their fixed output dimension and lack of pre-training knowledge.**
> > >
> > > **2. Generalization across task difficulty.**
> > > - As shown in Fig.3 and Table 6, the initial SFTed LLM struggles on challenging planning environments such as Sokoban and Sudoku. After SCOUT training on other symbolic tasks, the two task performance improves without direct optimization, suggesting that SCOUT learns reusable planning strategies rather than environment specific policies.
> > > In RubikCube, models trained only on lower difficulty settings (such as Rotation 3) successfully generalize to harder instances (Rotation 4,5), this demonstrates extrapolation to unseen task complexity.:
> > > |Trained on ↓\Tested on→|Rotation2|Rotation3|Rotation4|Rotation5|
> > > |-|-|-|-|-|
> > > |Rotation1|0.33|0.31|0.13|0.12|
> > > |Rotation2|\ |0.43|0.35|0.19|
> > > |Rotation3|\ |\ |0.58|0.45|
> > >
> > > **3. Generalization on different modalities (from text to visual).**
> > >  - To further evaluate transfer, we extend SCOUT to a visual POMDP setting. We use the model trained on Visual FrozenLake to evaluate on Visual Sokoban, which is also an unseen generalization setting. SCOUT achieves 0.31 success rate, outperforming both the instruct model (0.14) and standard RL training (0.20):
> > > |Tested on ↓\ Model→|Qwen2.5-VL-3B-it|with RL on Visual_Frozenlake| with SCOUT on Visual_Frozenlake|
> > > |-|----|------|--------|
> > > |VIsual_Sokoban|0.14|0.20|**0.31**|
> > >
> > > - This result suggests that the VLM with SCOUT recognizes and understands the provided image when achieving a very high score on its trained task, not just remembering the image.
> > > - This is also what small network scout could not achieve, because it is a specific task expert. The dimension of the action, state space are different in different tasks. Therefore, **a single-task trained small network scout could not transfer to another different environment.**
> > >
> > > ---
> > >
> > > In summary, our work provides the following key contributions:
> > > - By introducing SCOUT, we effectively mitigate the low exploration efficiency inherent in LLMs, enabling more efficient discovery in complex environments.
> > > - The SCOUT framework allows LLMs to achieve **SOTA** results on symbolic and spatial tasks that require heavy exploration. Our approach significantly outperforms previous leading methods, including **RAGEN (MMLS 2025), VAGEN’s state estimation (NeurIPS 2025), and SPA**.
> > > - We demonstrate that our solution does not merely excel in spatial reasoning and planning; it also provides robust generalization to broader **reasoning benchmarks** (e.g., GSM8K, ARC), proving that the learned strategies transfer beyond symbolic domains.
> > > - SCOUT is a highly generalizable framework. It is effective not only in state-based tasks but also in complex, continuous, and partially observable environments. Specifically, in visual POMDP settings, SCOUT achieved success rates of **0.89** on Visual_FrozenLake and **0.95** on Visual_Sokoban, substantially surpassing the current best VLM approach, **VAGEN (0.74 and 0.79, respectively)**.
> > >
> > >
> > > We hope these clarifications and additional results address your concerns. Please let us know if you have any further questions or require additional information.

---

### Decision · Program_Chairs · 2026-04-30

**Decision:**

Accept (regular)

**Comment:**

This paper explores improving LLM agents in unseen, non-linguistic environments, where standard language-based trial-and-error becomes prohibitively inefficient. The authors aim to consider a core issue: the mismatch between the exploration requirements of these environments and the computational constraints of large language models. The submission investigates a broad aspect of how to restructure the learning pipeline to make exploration tractable, and proposes SCOUT, a framework that decouples exploration and exploitation via lightweight “scout” models.

Reviewers agreed that the central idea is clear, well-motivated, and practically meaningful. The proposed approach of offloading exploration to small, efficient models and then distilling the resulting trajectories into an LLM is both simple and effective. Empirically, the paper demonstrates strong and consistent improvements across a range of symbolic and spatial tasks, with smaller models outperforming larger proprietary systems. In addition, the reported computational savings are substantial and well-supported by detailed analysis, highlighting the practical value of the approach. The framework also shows encouraging signs of generalization and stability in multi-task settings.

At the same time, reviewers raised several limitations. The evaluation is largely restricted to relatively simple or structured environments where the scout models already achieve strong performance, which raises questions about the necessity of distilling into an LLM rather than directly using the scout policy. There are also concerns about the reliance on the textualization step and how well the approach extends to more complex or noisy environments. Finally, some aspects of the experimental scope and presentation could be further strengthened.

Overall, this research's main result is that decoupling exploration from exploitation through sub-scale collaboration can substantially improve both the efficiency and effectiveness of LLM-based agents in unseen environments. Despite some limitations in scope and evaluation, the paper presents a clear idea, strong empirical evidence, and a practically impactful direction. My recommendation is to accept.